



# Quantifying nitrogen losses in oil palm plantations: models and challenges

**Lénaïc Pardon**[1], **Cécile Bessou**[1], **Nathalie Saint-Geours**[2], **Benoît Gabrielle**[3], **Ni'matul Khasanah**[4], **Jean-Pierre Caliman**[1,5], **Paul N. Nelson** [6]

[1]CIRAD, UPR Systèmes de pérennes, F-34398 Montpellier, France

emails : lenaic.pardon@cirad.fr; cecile.bessou@cirad.fr

[2]ITK, CEEI CAP ALPHA Avenue de l'Europe, 34830 Clapiers, France

email : nathalie.saint-geours@itk.fr

[3]AgroParisTech, INRA, UMR EcoSys, 78850 Thiverval-Grignon, France

email: benoit.gabrielle@agroparistech.fr

[4]World Agroforestry Centre (ICRAF), Southeast Asia Regional Programme, Bogor, Indonesia

email: j

[5]SMART Research Institute, Jl. Tengku Umar 19, Pekanbaru, 28112, Indonesia

email: j.p.caliman@sinarmas-agri.com

[6]College of Science and Engineering, James Cook University, Cairns 4878 Qld, Australia

email: paul.nelson@jcu.edu.au

*Correpondence to:* Cécile Bessou (cecile.bessou@cirad.fr)

**Key words:** Model testing, Oil palm, Nitrogen budget, Nitrogen losses

**Abstract.** Oil palm is the most rapidly expanding tropical perennial crop. Its cultivation raises environmental concerns, notably related to the use of nitrogen (N) fertilisers and associated pollution and greenhouse gas emissions. While numerous and diverse models exist to estimate N losses from agriculture, very few are available for tropical perennial crops. Moreover, there has been no critical analysis of the performances of existing models in the specific context of tropical perennial cropping systems. We assessed the capacity of 11 models and 29 sub-models to estimate N losses in a typical oil palm plantation over a 25-year-growth cycle, through leaching and runoff, and emissions of $NH_3$, $N_2$, $N_2O$, and $NO_x$. Estimates of total N losses were very variable, ranging from 21 to 139 kg N.ha$^{-1}$.yr$^{-1}$. On average, 31% of the losses occurred during the first three years of the cycle. Leaching comprised about 80% of the losses. Based on a comprehensive Morris sensitivity analysis, the most influential variables were soil clay content, rooting depth and oil palm N uptake. We also compared model estimates with published field measurements. Many challenges remain to model more accurately processes related to the peculiarities of perennial tropical crop systems such as oil palm.

## 1. Introduction

Oil palm is the most rapidly expanding tropical perennial crop. The area of land under oil palm, currently approximately 19 Mha, has been rising at 660,000 ha/year over the 2005-2014 period (FAOSTAT 2014) and is likely to continue rising until 2050 (Corley 2009). This increase raises significant environmental concerns..



Beside issues related to land-use changes and the oxidation of peat soils when establishing plantations, the cultivation of oil palm can generate adverse environmental impacts, in particular through the use of nitrogen (N) fertilisers. The latter are associated with pollution risks of ground and surface waters, and emissions of greenhouse gases (Choo et al., 2011; Comte et al., 2012; Corley and Tinker, 2008). An accurate estimation of N

losses from palm plantations is hence critical for reliable assessment of their environmental impacts. Models are necessary for such estimation, because comprehensive direct measurement of losses is prohibitively difficult and expensive.

While a number of models exist to estimate N losses from agricultural fields, they mostly pertain to temperate climate conditions and annual crops. Few models are available for tropical crops, and even fewer for perennial

tropical crops (Cannavo et al., 2008). Such models, in particular mechanistic ones, were primarily developed for research purposes, in order to simulate crop growth related to biogeochemical cycles and gain insight into the underlying processes. Nowadays, models are also widely used to estimate the emission of pollutants for the purpose of environmental assessment. Various models are used, from highly complex process-based models to more simple operational models such as empirical regressions. Despite some consensus and recommendations

regarding best practices for field emission modelling, notably within the framework of life cycle assessment (e.g. IPCC, 2006; EC ILCD, 2011), there has not been any comprehensive review and comparison of potentially useful models for environmental assessment. Moreover, various publications pinpointed the need for better adapted models in order to accurately model field emissions in the case of tropical crops (Basset-Mens et al., 2010; Bessou et al., 2013; Cerutti et al., 2013). In order to better target the needs for improving field emissions

modelling in oil palm plantations, there is a fundamental need to establish the potential applicability and pitfalls of state-of-the-art models regarding N cycling and losses in the conditions of oil palm plantations.

Most environmental impact assessment methods, such as life cycle assessment, model perennial systems similarly to annual ones. In these cases, the inventory data on the farming system and environmental fluxes are generally based on one productive year only, being the year of the investigation or the year for which data is

available (Bessou et al., 2013; Cerutti et al., 2013). Models of annual cropping systems do not account for differences in N cycling that occur during the cycle of perennial crops such as oil palm. Some key parameters, such as the length of the crop cycle, the immature and mature stages, inter-annual yield variations and other long-term eco-physiological processes, such as the delay between floret meristem initiation and fruit bunch harvest, are thus not accounted for. To improve the reliability and representativeness of model estimates to assess

environmental impacts of oil palm, there is therefore a need to better account for the spatio-temporal variability of both the agricultural practices and the eco-physiological responses of the stand throughout the perennial crop cycle (Bessou et al., 2013). Since most of these impacts hinge on N management and losses, modelling the N budget of palm plantations is a key area for improvement and is the focus of this work.

The objective of this work was to assess the capacity of existing models to estimate N losses in oil palm

plantations, while accounting for the peculiarities of oil palm plantations related to the N dynamics over the course of the growth cycle. It starts with a review of models that could be used for oil palm, details how they were selected, calibrated and run with relevant input data for a particular case study. Outputs from the models are then compared to each other and to previously reported field measurements. Key parameters are identified using



Morris sensitivity analysis. Finally, we discuss the relevance of existing models and the remaining challenges to adequately predict N fluxes in oil palm plantations.

## 2. Material and methods

### 2.1. Model selection and description

Among existing models, we first selected those that appeared most comprehensive and relevant. We then also selected partial models, in order to cover as much as possible various modelling approaches, and explore
potential complementarities between them. By "partial models" we mean models that simulate only one or a few N losses.

The selection criteria were, i) the possibility of estimating most of the N losses of the palm system; ii) applicability to the peculiarities of the oil palm system; and additionally, for partial models, iii) those most widely used in environmental assessments, e.g. EMEP (from European Environment Agency, 2013). In total, we
selected 11 comprehensive plus 5 partial models.

We compared models at two levels. At the first level the aim was to compare the 11 comprehensive models, to obtain an overview of their abilities to estimate the various N fluxes constituting the complete N budget of the plantations. The second level involved the partial models and aimed at better understanding the factors governing the variability of each type of N loss. Most of the 11 comprehensive models were actually a
compilation of sub-models. We hence included these sub-models in the second-level comparison, in addition to the 5 partial models originally selected. In total, 29 partial models, hereafter referred to as sub-models, were compared at this second level.

### 2.1.1 Description of comprehensive models

Following the definition of Passioura (1996), three of the comprehensive models were mechanistic, dynamic
models (WANULCAS from van Noordwijk et al., 2004; SNOOP from de Barros, 2012; APSIM from Huth et al., 2014). The others were simpler static models mainly based on empirical relationships (Mosier et al., 1998; NUTMON from Roy, 2005; IPCC 2006, from Eggleston et al., 2006; Banabas, 2007; Schmidt, 2007; Brockmann et al., 2014; Meier et al., 2014; Ecoinvent V3 from Nemecek et al., 2014). Other mechanistic models commonly used in crop modelling, such as DNDC (Li, 2007) and Century (Parton, 1996), were not adapted for oil palm
modelling and could not be used within our model comparison without a proper preliminary research and validation work, which fell beyond the scope of this work.

The mechanistic models were built or adapted explicitly for oil palm. The other models were developed or are mainly used for environmental assessment. Among the latter, some were explicitly built for oil palm or proposed parameters adaptable to oil palm (Banabas, Schmidt, Ecoinvent V3), some involved parameters potentially
adaptable to perennial crops (NUTMON, Brockmann, Meier 2014), while the others were designed to be used in a wide range of situations, without specific geographical or crop-related features (Mosier and IPCC 2006, which are often used in Life Cycle Inventories).

Most of the models took into account mineral and organic fertiliser inputs, some included symbiotic N fixation, and a few considered atmospheric deposition and non-symbiotic N fixation (Table 1). All models required



parameters related to soil, climate, and oil palm physiology, except for two (Mosier and IPCC 2006), which did not need any parameters other than the N input rates. Management parameters were mainly related to fertiliser application, i.e. amount, type and date of application of fertilisers. The splitting of application was considered in APSIM, SNOOP and WANULCAS, and the placement of the fertiliser was only taken into account in WANULCAS.

All models either used as input data or modelled the main internal fluxes of N. The most common fluxes were transfer from palms to soil, via mineralisation of N, in the residues left by the palms of the previous cycle and pruned fronds, followed by oil palm uptake and root turnover. The least considered fluxes were cycling of N through the other oil palm residues such as male inflorescences and frond bases, and uptake and recycling by legumes (in 5 models only).

Finally, the main losses modelled were leaching (all models), $N_2O$ emissions (10 models), and $NH_3$ volatilisation from fertilisers (9 models). $NO_x$ emissions and runoff were taken into account by fewer models (7 and 8 models, respectively). $N_2$ emissions, erosion, and $NH_3$ volatilisation from leaves were the least modelled losses. In some cases, several losses were modelled jointly and it was not possible to differentiate the contribution of each loss. For instance, erosion was always combined into the calculation of leaching and runoff, except for NUTMON,

which used the mechanistic erosion sub-model LAPSUS (Schoorl et al., 2002). However, we could not run LAPSUS since it required precise local parameters to run its digital terrain model component that were not available.

**TABLE 1. about here**

### 2.1.2. Description of sub-models

Each of the 29 sub-models modelled N losses from the soil-plant system via one of the following three types of pathways: loss via leaching and runoff (8 sub-models); loss by emission of $NH_3$, commonly referred to as volatilisation (9 sub-models); and loss by emission of the gaseous products of nitrification and denitrification: $N_2$, $N_2O$, and $NO_x$ (12 sub-models).

For the first pathway, being leaching and runoff, 8 sub-models were tested. Leaching concerned inorganic N

losses ($NO_3^-$, $NH_4^+$), whereas runoff included inorganic and organic N losses without specifying the dissolved or particulate form. Leaching was taken into account by all 8 sub-models. Runoff was calculated jointly with leaching in 2 sub-models (Mosier and IPCC 2006), and separately in modules of APSIM, SNOOP and WANULCAS. None of the 8 models calculated erosion losses. The Mosier and IPCC 2006 sub-models calculated losses as a linear function of N inputs via mineral and organic fertiliser applications and crop and

legume residues. Both used an emission factor of 30% of N inputs in our test conditions. Smaling (1993), SQCB-NO3 (Faist-Emmenegger et al., 2009) and Willigen (2000) used regressions and calculated losses taking into account N inputs, soil such as soil N organic content and soil clay content, climate such as annual rainfalls and some physiological parameters such as root depth and uptake rate. The input variables used depended on the sub-models. APSIM, SNOOP and WANULCAS used a soil N module coupled with a water budget module to

calculate the losses through leaching and runoff. In these three cases, a cascading layered approach was used to model the soil, and N transformation rates and water flows were calculated for each layer on a daily time step. The other 5 sub-models used a yearly time step.



For the second pathway, being volatilisation of $NH_3$, 9 sub-models were tested. They modelled $NH_3$ emissions from mineral and organic fertilisers, and 3 sub-models accounted for emissions from leaves. All sub-models

calculated the emissions from mineral fertiliser, except for Agrammon Group (2009), and 4 sub-models calculated the emissions from organic fertiliser. For emissions from leaves, Agrammon used a constant of 2 kg $N.ha^{-1}.yr^{-1}$, whereas EMEP 2009 (European Environment Agency, 2009) and EMEP 2013 (European Environment Agency, 2013) calculated these emissions jointly with emissions from mineral fertiliser. For emissions from organic and mineral fertilisers, the sub-models assumed linear relationships between fertiliser

application rate and N losses. The emission factors were modulated depending on the fertiliser type. For mineral fertilisers, emission factors ranged from 0 to 15% of N inputs for ammonium sulphate and 10 to 39% of N inputs for urea. For organic fertilisers, emission factors ranged from 20 to 35% of N inputs. For Mosier and IPCC 2006, emission factors differed only between mineral and organic fertilisers. In some sub-models, they were also modified by other parameters. For instance, in Bouwman et al. (2002a), the parameters were calculated taking

into account soil pH, soil temperature and cation exchange capacity; and in Agrammon, the emission factor was modified by factors specific to animal manure, e.g. manure from pigs versus cattle, application method, and not relevant for empty fruit bunches, the main organic fertiliser used in oil palm plantations.

For the third pathway, being gaseous losses of $N_2$ and N oxides, 12 sub-models were tested. $N_2O$ emissions were estimated by 8 sub-models. $NO_x$ emissions were estimated by 4 sub-models. $N_2$ emissions were estimated by 4

sub-models but calculated jointly with other gases, except for WANULCAS and APSIM. Mosier, IPCC 2006, Crutzen et al. (2008), EMEP 2013 and Nemecek et al. (2007) sub-models calculated losses as a linear function of N inputs, using fixed emission factors for $N_2O$, from 1 to 4% of N inputs, or $NO_x$ with 2.6% of N inputs in EMEP 2013. Meier 2012 also used a linear relationship, but with an emission factor that could be modified. However, its correction factors were applicable to annual crops under temperate climate and not here, e.g. impact

of tillage. Bouwman et al. (2002b), Shcherbak et al. (2014), and SimDen (Vinther and Hansen, 2004) sub-models used non-linear relationships between N inputs and N losses. Bouwman 2002b took into account various parameters for the calculation, mainly drainage, soil water content and C organic content. Shcherbak and SimDen took into account only N inputs and baseline emissions. APSIM and WANULCAS calculated the losses by the combination between a soil N module and a water budget module, plus a carbon module for APSIM.

**2.2. Model runs and sensitivity analysis**

**2.2.1. Model calibration and input data**

Oil palm plantations are usually established for a growth cycle of approximately 25 years. Palms are planted as seedlings and the plantation is considered immature until about 5 years of age, when the palm canopy closes and the plantation is considered mature. Harvesting of fresh fruit bunches starts after about 2-3 years. The models

were run over the whole cycle, including changes in management inputs and output yields between immature and mature phases. We considered replanting after a previous oil palm growth cycle. Potential impacts of land use change on initial conditions were hence not considered. However, when possible, the initial decomposing biomass due to felling of previous palms was included in the models.

In order to compare the models, we kept calibration parameters and input variables consistent across models as

much as possible. However, all models did not need the same type of parameters and input data. In particular, for





some static models input variables were initially fixed and could be considered as calibrated parameters based on expert knowledge. For instance, NUTMON and Ecoinvent V3 needed the oil palm uptake rate as an input value, but Schmidt and APSIM used their own calculations for uptake.

We considered one ha of plantation located in the Sumatra region of Riau. For climate during this period, the dataset contained daily rain being of 2407 mm.yr$^{-1}$ on average, temperature and solar radiation. As the dataset was only 16 years long, from 1998 to 2013, we had to repeat an average year to complete the last 9 years of the simulation. The soil was a typical Ultisol, with four layers (0-5, 5-15, 15-30, and 30-100 cm). The main characteristics, averaged over the upper 30 cm, were bulk density (1.4 metric tonne.m$^{-3}$), drainage (good), clay content (31%), initial organic C content (1.65%, i.e. 0.0165 g.g$^{-1}$), initial organic N content (5.5 t.ha$^{-1}$), pH (4.5), 195 and rate of soil organic N mineralisation (1.6% per year) (USDA soil taxonomy, 1999; Khasanah et al., 2015; Corley and Tinker, 2003; Roy, 2005).

Regarding management input variables, we used a set of values representing a standard average industrial plantation (Pardon et al., 2016). These values were consistent and based on a comprehensive review of available measurements. For oil palm the main peculiarities were the yield (25 t of fresh fruit bunches.ha$^{-1}$.yr$^{-1}$ after ten 200 years, i.e. 73 kg N.ha$^{-1}$.yr$^{-1}$), the uptake (222 kg N.ha$^{-1}$.yr$^{-1}$ after ten years), and the depth where most of the active roots are found (set at 1m). For the management of the field, the input variables were the slope (0°), planting density (135 palms.ha$^{-1}$), presence of a legume cover sown at the beginning of the cycle (e.g. *Pueraria phaseoloides* or *Mucuna bracteata*), and presence of the biomass of felled palms from the previous growth cycle (550 kg N.ha-1, corresponding to the above- and below-ground biomass of felled palms). For fertiliser, the 205 application of mineral fertiliser increased from 25 kg N.ha$^{-1}$.yr$^{-1}$ the first year up to 100 kg N.ha$^{-1}$.yr$^{-1}$ after the fifth year. It was assumed to be 25% of urea and 75% of ammonium sulphate. Organic fertiliser, i.e. empty fruit bunches, was applied around the palms for the first two years as a typically used rate of 184 kg N.ha$^{-1}$.yr$^{-1}$. That amount, over two years, corresponds to the amount of empty fruit bunches generated from one hectare over 25 years, assuming a yield of 25 t of fresh fruit bunches.ha$^{-1}$.yr$^{-1}$. Atmospheric deposition of N through rain was set 210 at 18 kg N.ha$^{-1}$.yr$^{-1}$. Biological N fixation by the legume cover was set as 635 kg N.ha$^{-1}$ fixed over the first 7 years, and released to the soil during the same period. The release of N through the decomposition of the organic residues from palms was set at an annual average of 108 kg N.ha$^{-1}$.yr$^{-1}$ going to the soil. These residues correspond to fronds and some inflorescences which are regularly pruned, frond bases naturally falling, and dead roots.

For model comparison, we calculated the annual estimated losses, considering the relative contributions of leaching, runoff and erosion; $NH_3$ volatilisation; and $N_2$, $N_2O$, and $NO_x$ emissions. Besides inter-model comparison, we also compared the simulated losses with previously reviewed measurements from the literature (Pardon et al., 2016). Most of the models are static ones and do not account for variations in processes during the crop cycle. In order to model the whole cycle, we ran these models on a yearly basis accounting for annual 220 changes in some input variables from the scenario, such as fertiliser application rates, biological N fixation, crop N uptake, N exported in fresh fruit bunches, temperature, rainfalls, etc. One model, i.e. SNOOP, simulates specific years of the crop cycle one by one, using a daily time step. For this model, the calculation was repeated 25 times taking into account the year-to-year changes. The other models were built to simulate the whole growth





cycle with a daily time step, as for WANULCAS and APSIM, or with a yearly time step, as for Banabas and
Schmidt.

For the sub-model comparisons, we compared separately the three groups of sub-models: 1) leaching, runoff,
erosion; 2) $NH_3$ volatilisation; 3) $N_2$, $N_2O$, and $NO_x$ emissions. For these comparisons, we used the same input
data and the same calibration as for the previous one.

We compared the magnitude of the losses estimated by the various sub-models, and when possible, we also
identified the contribution of the various N input sources to the losses estimated, i.e. the influence of mineral
fertiliser, organic fertiliser, biological N fixation, plant residues and atmospheric depositions.

**2.2.2. Sensitivity analysis**

Sensitivity analysis investigates how the uncertainty of a model output can be apportioned to different sources of
uncertainty in the model inputs (Saltelli et al., 2008). Sensitivity analysis aims at ranking sources of uncertainty
according to their influence on the model outputs, which helps to identify inputs that should be better scrutinised
in order to reduce the uncertainty in model outputs.

We conducted a Morris sensitivity analysis (Morris, 1991) for the three groups of sub-models in order to identify
the input variables having most effect on the magnitude of the losses. We used RStudio software to code and run
the models (R Development Core Team, 2010), and the "morris" function from the "sensitivity" package version
1.11.1. SNOOP, APSIM and WANULCAS were not included in the sensitivity analysis as the source codes were
either not accessible or not directly programmable without adapting the model structure to run the sensitivity
analysis, which fell beyond the scope of this study.

Each model used $n$ input variables. For each input variable $X_i$ ($i \in [1; n]$), we defined a nominal, minimum, and
maximum value. For climate, soil, oil palm characteristics, and N input fluxes, the ranges were determined based
on references. For emission factors and other parameters, the ranges were given by the sub-models, e.g. in IPCC
2006, or were varied from -90% to +90% of the nominal value if the sub-models did not specify ranges. The
ranges and references are listed in Table SM1 in Supplementary material. For the analysis, each range was
normalised between 0 and 1 and then split into 5 levels by the "morris" function.

The Morris sensitivity analysis technique belongs to the class of "One-at-a-time" sampling designs. For each
model, we carried out 400*($n + 1$) runs, with each set of $n + 1$ runs called a "trajectory". For each trajectory, an
initial run was carried out in which all input variables were set to a random level out of the 5 possible. For the
second run, one variable $X_1$ was changed to another random level different from the initial one, and the
difference in output between the first and second runs was recorded. That difference, divided by the normalised
change in input level, is called an "elementary effect" of $X_1$. For the third run another variable $X_2$ was changed,
keeping all other input variable values the same as in the second run. The elementary effect of $X_2$ was recorded,
and so on, until the $(n + 1)th$ run. Each trajectory was initiated using a new random set of input variable values,
and each trajectory generated one elementary effect for each $X_i$.

Then, following Morris's method, we calculated two sensitivity indices for each variable $X_i$: the mean of
absolute values of the 400 elementary effects ($\mu_i^*$), being the mean influence on the output when the input varies




in its min/max range, and their standard deviation ($\sigma_i$). The higher $\mu_i^*$ was, the more influential was the variable $X_i$. The higher $\sigma_i$ was, the more important was the interaction between the variable $X_i$ and the other input variables in the model, or the influence of $X_i$ was nonlinear. The mean of their absolute values of the elementary effect ($\mu_i^*$) was used rather than the mean of the actual values ($\mu_i$), because the effects could be positive or negative.

**3. Results**

**3.1. Comparison of the 11 comprehensive models**

Estimations of total losses of N were very variable, with an average of 77 kg N.ha$^{-1}$.yr$^{-1}$, ranging from 21 to 139 kg N.ha$^{-1}$.yr$^{-1}$ (Figure 1.a). Two main factors influenced this variability: some pathways were not taken into account by all the models (see Table 1); and estimates of leaching, runoff and erosion, which greatly contributed

to the total losses, were particularly variable across models.

According to the models, the leaching and runoff pathway was the most important of the three, with an average loss of 61 kg N.ha$^{-1}$.yr$^{-1}$, i.e. about 80% of the losses, ranging from -12 to 135 kg N.ha$^{-1}$.yr$^{-1}$. Negative leaching loss was estimated with NUTMON after the sixth year, when oil palm N uptake exceeded 160 kg N.ha$^{-1}$.yr$^{-1}$. NH$_3$ volatilisation was the next most important pathway with 11 kg N.ha$^{-1}$.yr$^{-1}$ on average, ranging from 5 to 13.

Emissions of N$_2$, N$_2$O, and NO$_x$ had the lowest magnitude, but considerable variability, with 6 kg N.ha$^{-1}$.yr$^{-1}$ on average, ranging from 0 to 19 kg N.ha$^{-1}$.yr$^{-1}$.

According to the models, losses varied substantially along the growth cycle. On average, 31% of the losses occurred during the immature period, which represents 12% of the cycle duration (Figure 1.b). Most of the models simulated maximum losses near the beginning of the cycle. The magnitude of this peak was very

variable, up to 738 kg N.ha$^{-1}$.yr$^{-1}$ for Schmidt. Its timing in the cycle depended on the model, occurring for instance during the first, second, or fourth year for Ecoinvent V3, IPCC 2006 and APSIM, respectively (Figure 2: for clarity, only four examples are shown, to illustrate the variability of the results). This high loss of N near the start of the cycle was due to the large amount of N entering the soil during this time, in the felled palms from the previous cycle, the spreading of empty fruit bunches, and biological N fixation. The high

variability in the magnitude and timing of the peak was due to differences in modelling approaches, especially the inclusion or otherwise of various N inputs and internal fluxes.

**FIGURE 1. about here**

**FIGURE 2. about here**

**3.2. Comparison of the 29 sub-models**

**3.2.1. Losses through leaching and runoff**

For this pathway, 8 sub-models were tested (Figure 3), which were all sub-models integrated in the comprehensive models. There were no stand-alone models focusing on this pathway. Models from Banabas, Schmidt and Meier 2014 were not included in this comparison, because they did not use specific sub-models but



calculated leaching, runoff and erosion as the surplus of the N budget. The average loss estimate of the 8 sub-
models was 59 kg N.ha$^{-1}$.yr$^{-1}$, with a -12 to 135 kg N.ha$^{-1}$.yr$^{-1}$ range.

**FIGURE 3. about here**

All 8 sub-models considered leaching. Five models considered runoff, but this flux was very low, i.e.
< 0.06 kg N.ha$^{-1}$.yr$^{-1}$, due to the assumption of a zero field slope. None of these models considered erosion.
Therefore, the fluxes calculated for this pathway could be considered as leaching losses, and their variability
mainly hinged on the way leaching was modelled. In comparison, field measurements of this pathway type range
from 3.5 to 55.8 kg N.ha$^{-1}$.yr$^{-1}$ (Figure 4).

**FIGURE 4. about here**

Without accounting for N inputs via empty fruit bunches application, atmospheric deposition and biological N
fixation, the average annual losses were estimated at 26 kg N.ha$^{-1}$.yr$^{-1}$. There was a substantial variation between
sub-models, which spanned an overall range of -17 to 60 kg N.ha$^{-1}$.yr$^{-1}$ (mean of 6 sub-models). When empty
fruit bunches application was taken into account, the losses increased by an average of 3 kg N.ha$^{-1}$.yr$^{-1}$ (mean of
5 sub-models). When biological N fixation was taken into account, the losses increased by an average of
18 kg N.ha$^{-1}$.yr$^{-1}$ (mean of 2 sub-models).

In terms of temporal patterns (Figure SM1), APSIM estimated peak losses through leaching and runoff of up to
251 kg N.ha$^{-1}$ in the fourth year, when biological N fixation was taken into account. The peak losses through
leaching estimated by SQCB-NO3 more than doubled (up to 103 kg N.ha$^{-1}$) when empty fruit bunches
application was taken into account. This peak of losses through leaching at the beginning of the cycle has also
been observed in field measurements (Pardon et al., 2016).

In terms of spatial patterns, WANULCAS calculated that, of the 135 kg N.ha$^{-1}$.yr$^{-1}$ lost through leaching, about
88 came from the weeded circle surrounding the palm stem, where the mineral and organic fertilisers were
applied; and about 31 originated from the windrow where the trunks from the previous palms were left.

### 3.2.2. NH$_3$ volatilisation

For this pathway, 9 sub-models were tested (Figure 5). In this comparison, 2 sub-models were partial models not
used in the 11 comprehensive models (EMEP 2013 and Bouwman 2002a). Two sub-models were used by several
comprehensive models: Asman (1992) was used by Ecoinvent V3 and Meier 2014, and Agrammon was used by
Ecoinvent V3 and Brockmann. Modelled estimates averaged 10.0 kg N.ha$^{-1}$.yr$^{-1}$, with a range of 5.4 to
18.6 kg N.ha$^{-1}$.yr$^{-1}$.

**FIGURE 5. about here**

Whenever possible, we differentiated the influence of mineral fertiliser, empty fruit bunches and leaves on the
emissions. The average emissions from mineral fertiliser were estimated at 9.2 kg N.ha$^{-1}$.yr$^{-1}$ (mean of 8 sub-
models). The emission factors for urea and ammonium sulphate differed considerably between models, ranging
from 10 to 39% and 1.1 to 15%, respectively. However, in several cases these differences compensated for each
other when total emissions from mineral fertiliser were calculated. For instance, emissions calculated using the



models of Schmidt and Asman were close, with 8.4 and 9.1 kg N.ha$^{-1}$.yr$^{-1}$, respectively, whereas their emission factors were very different, being 30 and 2% in Schmidt, 15 and 8% in Asman; for urea and ammonium sulphate, respectively. The average emissions from empty fruit bunches were estimated at 3.7 kg N.ha$^{-1}$.yr$^{-1}$ (mean of 4 sub-models). But these estimates were done with emission factors more adapted to animal manure than to empty fruit bunches. The emissions from leaves were estimated separately only by Agrammon, with a constant rate set by definition in the model at 2 kg.ha$^{-1}$.yr$^{-1}$. For comparison, field measurements of losses as $NH_3$ range from 0.1

to 42 kg N.ha$^{-1}$.yr$^{-1}$ (Figure 4).

In terms of temporal patterns, only the sub-models considering emissions from empty fruit bunches presented a peak which occurred over the first two years.

### 3.2.3. $N_2O$, $N_2$, $NO_x$ emissions

For this pathway, 12 sub-models were tested (Figure 6). Three of these sub-models were partial models not used

in the 11 comprehensive models (Crutzen, EMEP 2013 and Shcherbak). Four sub-models were used in several comprehensive models: Nemecek 2007 was used in Ecoinvent V3 and Brockmann; and IPCC 2006 was used in Schmidt, Ecoinvent V3, Meier 2014 and Brockmann. The average estimate of combined $N_2$, $N_2O$ and $NO_x$ emissions was 5.2 kg N.ha$^{-1}$.yr$^{-1}$, with a wide range from 0 to 19.1 kg N.ha$^{-1}$.yr$^{-1}$. This wide range could be explained partly because some sub-models estimated only $N_2O$ or $NO_x$, while others calculated two or three of

these gases jointly. Therefore, we did comparisons for $N_2O$ and $NO_x$ separately, in order to better understand the variability of the results. Emissions of $N_2$ were always calculated jointly with another gas, except for WANULCAS and APSIM. When possible, we also determined the influence of mineral fertiliser, empty fruit bunches, biological N fixation, plant residues and soil inorganic N on emissions.

**FIGURE 6. about here**


For $N_2O$, the average estimate of the outputs was 3.4 kg N.ha$^{-1}$.yr$^{-1}$, ranging from 0.3 to 7 kg N.ha$^{-1}$.yr$^{-1}$ across 8 sub-models (Figure 7). The average contributions were estimated at 2.0 kg N.ha$^{-1}$.yr$^{-1}$ for mineral fertiliser (mean of 6 sub-models), 0.8 for empty fruit bunches (mean of 4 sub-models), 0.5 for biological N fixation (mean of 3 sub-models), 1.6 for plant residues (mean of 3 sub-models), and 1.6 for soil inorganic N (1 sub-model). In this

range of results, it was difficult to identify the most suitable models. For instance, Bouwman 2002b seemed relevant as it used a climate parameter for sub-tropical context. Shcherbak seemed relevant for oil palm management as it calculated losses as a non-linear function of N inputs, which avoids overestimating emissions when mineral fertiliser inputs were less than 150 kg N.ha$^{-1}$.yr$^{-1}$. Yet, the results were very different, being the highest for the former, with 7 kg N.ha$^{-1}$.yr$^{-1}$, and one of the lowest for the latter, with 0.8 kg N.ha$^{-1}$.yr$^{-1}$. For $NO_x$,

the average estimate of the outputs was 1.4 kg N.ha$^{-1}$.yr.$^{-1}$, ranging from 0.3 to 2.4 kg N.ha$^{-1}$.yr$^{-1}$ across 4 sub-models (Figure 8). In comparison, measurement-based estimates of the losses as $N_2O$ range from 0.01 to 7.3 kg N.ha$^{-1}$.yr$^{-1}$ (Figure 4).

**FIGURE 7. about here**

**FIGURE 8. about here**





In terms of temporal patterns (Figure SM2), the sub-models that included mineral fertiliser inputs only did not
show any peak of emissions over the crop cycle, e.g. in Bouwman 2002b, whereas the ones taking into account
at least one other N input, such as felled palms, empty fruit bunches, biological N fixation, showed a peak during
the immature period, e.g. in Crutzen and APSIM. In field measurements, higher levels of losses through $N_2O$
have also been observed at the beginning of the cycle (Pardon et al., 2016). With some sub-models the peak

occurred during the first three years of the cycle, e.g. at 10 kg $N.ha^{-1}.yr^{-1}$ in the second and third years in
Crutzen, but in APSIM it occurred later, at 9 kg $N.ha^{-1}.yr^{-1}$ in the fourth year.

### 3.3. Sensitivity analysis

For the leaching and runoff pathway, 5 out of 8 sub-models were tested (Figure 9). None of these sub-models
took erosion into account. We therefore did not test the influence of slope. On average for the 5 sub-models, the

most influential input variables were clay content, rooting depth, oil palm N uptake, and the IPCC emission
factor, resulting in values of $\mu^* > 200$ kg $N.ha^{-1}.yr^{-1}$. For clay content, rooting depth and oil palm N uptake, there
were also high nonlinearities and/or interactions with other variables, with $\sigma > 250$ kg $N.ha^{-1}.yr^{-1}$. In the case of
clay content, the variability was substantial. It was very influential for SQCB-NO3 and Willigen, with $\mu^* > 395$
kg $N.ha^{-1}.yr^{-1}$ and $\sigma > 1200$ kg $N.ha^{-1}.yr^{-1}$, but had no influence for Smaling, with $\mu^*$ and $\sigma$ were zero, which was

not sensitive to clay content when it was less than 35%. Nitrogen inputs, through mineral fertiliser application,
empty fruit bunches application, and biological N fixation, and rainfall, had lower mean influence and lower
nonlinearities and/or interaction indices, $\mu^*$ ranging from 64 to 110 kg $N.ha^{-1}.yr^{-1}$ and $\sigma$ ranging from 40 to 141
kg $N.ha^{-1}.yr^{-1}$. Other input variables related to soil characteristics, such as carbon content and bulk density, had
lower mean influences with $\mu^* < 45$ kg $N.ha^{-1}.yr^{-1}$.

**FIGURE 9. about here**

For $NH_3$ volatilisation, 7 out of 9 sub-models were tested (Figure 10). The influences of input variables were
lower for this pathway than for leaching and runoff, with $\mu^* < 80$ kg $N.ha^{-1}.yr^{-1}$ and $\sigma < 35$ kg $N.ha^{-1}.yr^{-1}$. For the
7 sub-models, the mean influences of variables related to organic fertiliser, i.e. emission factor and rate of
application, were on average higher than for mineral fertiliser, i.e. emission factor, rate of application and urea

rate in fertiliser applied, with $\mu^*$ being 38 to 78 kg $N.ha^{-1}.yr^{-1}$ and 12 to 32 kg $N.ha^{-1}.yr^{-1}$; respectively. The
interaction indices were also higher for organic fertilisers than for mineral fertilisers. Temperature and soil pH
were less influential with $\mu^* < 2$ kg $N.ha^{-1}.yr^{-1}$.

**FIGURE 10. about here**

For $N_2$, $N_2O$ and $NO_x$ emissions, 7 out of 12 sub-models were tested (Figure 11). The influences of input

variables were lower for this pathway type than for the other two, with $\mu^* < 44$ kg $N.ha^{-1}.yr^{-1}$ and
$\sigma < 19$ kg $N.ha^{-1}.yr^{-1}$. However, the mineral fertiliser rate had a very high mean influence compared to the other
pathway types, being $\mu^*$: 44 kg $N.ha^{-1}.yr^{-1}$, because one sub-model was very sensitive to the fertiliser application
rate, i.e. $\mu^*$: 283 kg $N.ha^{-1}.yr^{-1}$ for Shcherbak. Most of the N inputs had a lower mean influence on emissions
than emission factors, except for biological N fixation.

**FIGURE 11. about here**



Across the three pathways, i.e. 19 sub-models, the 5 most influential variables were related to leaching and runoff losses (Figure 12). These variables, which had $\mu^*$ greater than 100 kg N.ha$^{-1}$.yr$^{-1}$, were clay content, oil palm rooting depth, oil palm N uptake, and emission factors of IPCC 2006 and Mosier. Their interaction indices were also very high, except for the two emission factors. Mineral and organic fertiliser application rates and

biological N fixation were the only input variables not specific to one pathway but used to simulate losses in all the three pathways. Soil pH, temperature, and other N inputs in soil such as atmospheric N deposition, residues of legume and oil palm, had lower influences on losses.

**FIGURE 12. about here**

**4. Discussion**

**4.1. Relevance of model comparisons and flux estimates**

The model comparisons brought to light large variations between models estimating N losses from oil palm plantations. This variability concerned both the structures and outputs of the models. The comparison included process-based or regression-based, yearly or daily time-step, and more or less comprehensive models. We may assume that other models exist, which we could not access or calibrate, but those tested very likely provide a

representative sample of modelling possibilities for the N budget in oil palm plantations. Some model runs fell beyond their validity domains, especially regression-based models for leaching. As the study did not aim to validate the robustness of the models, we retained these outputs, and they helped highlight key fluxes and uncertainties. Further modelling work across contrasting plantation situations might be worthwhile to further test the validity of the models. In particular, nutrient, water or disease stresses or the impact of the previous land use

may critically influence the overall crop development and associated N budget.

There was an approximate 7-fold difference between the lowest and the highest overall N loss estimates. In order to investigate the plausibility of these model estimates, we used a simple budget approach. Assuming that soil N content remained constant over the cycle, N inputs would equal N exported in fresh fruit bunches plus the increase in N stock in palms plus N lost. The assumption of constant soil N appears reasonable because soil N is

closely related to soil C, and soil C stocks in plantations on mineral soil have been shown to be fairly constant over the cycle, especially when oil palm does not replace forest (Smith et al., 2012; Frazão et al., 2013; Khasanah et al., 2015). In our scenario based on measured values (Pardon et al., 2016), average N inputs, N exported and N stored in palms were 156, 60 and 22 kg N.ha$^{-1}$.yr$^{-1}$, respectively. Therefore, with a constant soil N stock over the cycle, N losses should be of about 74 kg N.ha$^{-1}$.yr$^{-1}$.

Based on this plausible estimate of 74 kg N.ha$^{-1}$.yr$^{-1}$, it was possible to identify three groups among comprehensive models: models likely to underestimate losses (IPCC, Mosier, Ecoinvent V3, NUTMON), models likely to overestimate losses (SNOOP, WANULCAS), and models likely to estimate a plausible amount of losses (Banabas, Meier 2014, Brockmann, APSIM, Schmidt).

Underestimates may be due to underestimation of leaching losses. This was particularly clear for SQCB-NO3

and NUTMON, which used regressions not adapted to the high oil palm N uptake. This even resulted in negative leaching losses for the latter. However, IPCC, Mosier and SQCB-NO3, estimated leaching losses within the range of measured losses through leaching, runoff and erosion of 3.5 to 55.8 kg N.ha$^{-1}$.yr$^{-1}$ (Figure 4). All models





seemed to underestimate $NH_3$ volatilisation compared with measured values (Figure 4). However, this was due to the fact that the higher measured value of 42 kg $N.ha^{-1}.yr^{-1}$ was for mineral fertiliser applications of solely

urea, whereas the rate of urea in our scenario was 25% of mineral fertiliser. For IPCC, Mosier and SQCB-NO3, underestimation may also be explained by the fact that none of them were complete budgets. They accounted neither for all gaseous emissions, such as emissions of $N_2$, nor for all inputs, such as atmospheric deposition.

Overestimates of losses were related to overestimation of leaching losses, which were very high for both WANULCAS and SNOOP. This overestimation could result from the interactions occurring between modules in

process-based models, e.g. between zoning and N inputs in WANULCAS. However, more work would be necessary to better understand how the structure of the model could lead to overestimates of leaching.

For estimates closer to the plausible 74 kg $N.ha^{-1}.yr^{-1}$, the results hide several cases. APSIM estimated a plausible overall loss of 84 kg $N.ha^{-1}.yr^{-1}$, but leaching seemed overestimated compared to measured values. This was very likely because some other fluxes were not taken into account, such as $NH_3$ volatilisation and N input through

empty fruit bunches. Similarly, Meier 2014 and Brockmann had plausible overall loss estimates, but leaching losses seemed overestimated, while neither $N_2$ emissions, nor N input through biological N fixation, were taken into account. Schmidt and Banabas estimates seemed to be plausible and they accounted for most of the fluxes. Modelled $N_2O$ emissions were similar to field measurements, although the minimum modelled emissions were still higher than the minimum losses measured in the field.

Some notable patterns differentiated process-based versus regression-based models and more comprehensive versus less comprehensive models. The process-based models tended to predict higher overall losses and appeared to overestimate leaching losses. The less comprehensive models either seemed to underestimate overall losses, or tended to overestimate leaching losses, which counteracted missing fluxes. Regarding leaching losses, the process-based models produced similar estimates those that deduced these losses from the total balance.

Process-based models have the advantage of being able to simulate the impact of management practices, such as timing, splitting and placement of fertilisers. They also take into account other related processes impacting upon the N cycle, such as carbon cycling, plant growth and water cycling. However such models need more data, e.g. related to soil characteristics, the interactions between modules may generate unexpected behaviors, e.g. for leaching estimation, and they are generally not easily handled by non-experts. On the other hand, simple models,

such as IPCC and Mosier, could provide plausible results if they were more comprehensive, without requiring a lot of data. However they cannot take into account peculiarities of oil palm or the effects of management practices. One path forward is the development of simple models, such as agro-ecological indicator based on the Indigo© concept (Girardin et al., 1999). These indicators are designed to be used easily while taking into account crop system specificities such as management practices.

**4.2. Challenges for modelling the N budget in oil palm plantations**

We identified two important challenges for better modelling the N cycle in oil palm plantations: 1) to model most of the N inputs and losses while accounting for the whole cycle, and 2) to model particular processes more accurately by accounting for peculiarities of the oil palm system (Table 2).



Given the changes in N dynamics, management practices, and N losses through the crop cycle, it is important for models to be built in a way that accounts for the whole growth cycle. In particular, the immature phase is an important period to consider, as 31% of the losses occurred during this phase according to the models. Measurements in the field have also shown losses to reach peak values during this phase (Pardon et al., 2016). This period involves large inputs through the felled palms, empty fruit bunch application and biological N fixation, with complex associated N dynamics in groundcover, litter and soil. Regarding input fluxes, besides the fertiliser applications, it seems important to also account for biological N fixation and atmospheric deposition since their contributions to the N budget were not negligible. Internal fluxes, such as decomposition of felled palms and residues of oil palm and groundcover, are among the largest fluxes in the oil palm system and their influence on N dynamics is substantial (Pardon et al., 2016). In the case of a new planting, the impacts of land use change and land clearing might also need to be further investigated to better quantify the input fluxes due to decomposition as well as the influence of transitional imbalance state of the agroecosystem on N loss pathways.

For N losses, further model development is also needed to complete the N balance. First, it would be worthwhile to model erosion without requiring too many and complex input data, while accounting for change in erosion risk through the crop cycle and effects of erosion control practices. Erosion was not modelled independently in many of the reviewed models. Further, $NH_3$ emissions from leaves could easily be included. Finally, $N_2O$, $NO_x$ and $N_2$ should be modelled in a more comprehensive and systematic way. In particular, $N_2O$ emissions, and thus presumably $NO_x$ and $N_2$ emissions, have high spatial and temporal variability (Ishizuka et al., 2005), related not only to fertiliser application parameters. The time resolution of $N_2O$ field measurements has proven to influence significantly the total amount of recorded $N_2O$ emissions (Bouwman et al. 2002b). It is hence paramount to model those N losses accounting for the changes in driving parameters over the whole crop cycle.

Finally, losses should not be calculated jointly if the objective is to perform environmental impact assessment and identify practices most likely to reduce the losses. Indeed, different N fluxes may lead to different N pollution risks. N losses through erosion, runoff or leaching do not end up in the same environmental compartments, e.g. surface water versus ground water. They hence do not contribute in the same way to potential environmental impacts such as eutrophication. For the purpose of environmental assessment, models should hence be as comprehensive and detailed as possible.

The second challenge is to better model some of the key N cycling processes while accounting for peculiarities of the oil palm system. For internal fluxes, to better model the interaction between legumes and soil N dynamics is an important challenge, as the actual role of legumes during the immature period is complex and not fully understood. Indeed, legumes have the capacity to regulate their N provision, by fostering N fixation or N uptake, depending on the nitrate content of the soil (Pipai, 2014; Giller and Fairhurst, 2003). They may contribute to reducing N losses through immobilisation or to enhancing N losses through N fixation and release.

Reducing the uncertainty of leaching modelling is an important challenge, as about 80% of the total losses came from leaching, according to the models, and results were very variable across models. Models should be better adapted to the oil palm system, as some regression models were out of their validity domain. The sensitivity analysis also showed that the most influential variables, upon which research should focus, were soil clay content, oil palm rooting depth and oil palm N uptake. In order to take into account the influence of management





practices on internal fluxes and losses, it would be necessary to use a daily step approach and to favour modelling approaches that incorporates spatial heterogeneity, as in WANULCAS. With a daily step approach it is possible to account for timing or splitting of N fertiliser application, and accounting for spatial heterogeneity

means placement of mineral fertiliser or empty fruit bunches can be assessed. For gaseous losses, emission factors could be adapted to the oil palm system, as all of them, i.e. for $NH_3$ or $N_2O/NO_x$ fluxes, were based on data from temperate areas on mineral soils, including mostly animal manure as reference for organic fertilisers. On a general note, more field measurements and model development are needed to account for the peculiarities of palm plantation management on peat soil. Plantation area on peat soil is substantial and potentially spreading,

notably in Indonesia (Austin et al., 2015). Those plantations require specific management, including complex drainage systems, and may have severe pollution risks, notably leaching, that are not yet properly accounted for in currently used models, e.g. IPCC 2006.

**TABLE 2. about here**

**4.3. Implications for management**

The main levers managers have for reducing losses are at the input level, including fertiliser management, but also the handling of the immature phase. To manage fertiliser inputs, managers need to know economic response, which is the main driver of practices, and environmental response, to type, rate, timing and placement. Then they can decide the optimum fertiliser management practices. Models that include both N losses and fresh fruit bunch production in relation to this management can provide the information needed.

The model comparison showed the importance of the immature phase for losses and suggested field research and modelling approaches to improve understanding and loss estimates. There are also implications for management during this phase. Light, water and N are not being fully used by the young palms, as their canopies and root systems do not completely exploit the field. Thus, in the current systems, the combination of high inputs and small capture by the growing oil palms in the immature period causes negative environmental impacts. There are

two possible approaches for reducing these losses. One is to reduce the inputs. For instance, it might be better to plant a non-legume cover crop and to manage N supply to the palms with fertiliser only. An alternative approach would be to cultivate another crop during this phase, which would use the surplus N and either export it in product or take it up in biomass that would decompose later. For instance, fast growing trees like balsa; trunks could be harvested after 5 years and exported, whilst leaving some branches, leaves and roots to decompose.

Re-planting systems also exist that allow for combining old and young palm trees in the same plantation block. The advantage can be both economic and agro-ecologic as the immature phase actually becomes productive thanks to the remaining old palm trees and the nutrient cycling potentially more competitive. However, there is still limited data available to quantify and model the potential competition and adapt fertiliser management. Moreover, potential reduction in N losses should not come at the cost of increased emissions of herbicides,

which may be used to kill the old palm trees without damaging the newly planted ones.

From the environmental point of view, it is also important to consider fertiliser management and N losses within a wider system. First, fertilisers encompass residues from the mill, whose cost/benefit influence on the final balance should be considered over the whole life cycle, i.e. including production, transport, or avoided impact



through the substitution of synthetic fertilisers, etc. This can be done using life cycle assessments. Second, the
carbon balance, i.e. the balance of carbon sequestration and release, is closely coupled to the N balance. Thus
models that include both cycles are advantageous for evaluating environmental impacts.

**5. Conclusion**

N losses are a major concern for environmental impact, and management to reduce N losses and costs is critical
in oil palm cultivation. Modelling N losses is crucial because it is the only feasible way to predict the type and
magnitude of losses, and thus to assess how improved management practices might reduce losses. Our study
showed that there were considerable differences between existing models, in terms of model structure,
comprehensiveness and outputs. The models likely to generate N loss estimates close to reality seem to be those
that are most comprehensive and take into account the main oil palm peculiarities, irrespective of time step.
However, in order to be useful for managers, a precise modelling of the impact of management practices on all N
may require the use of a daily time step or the modelling of spatial heterogeneity. The main challenges are to
better understand and model losses through leaching, and to account for most of the N inputs and outputs.
Leaching is the main loss pathway and is particularly high during the immature phase when inputs are high due
to decomposition of felled palm and fixation by legumes. Data is still needed to better understand temporal and
spatial variability of other losses as well, such as $N_2$, $N_2O$ and $NO_x$ emissions, in the context of oil palm. These
improvements could allow managers to evaluate the economic and environmental impacts of changes in
management, such as for instance, modifying fertiliser inputs or the plant cover type during the immature phase.

**Acknowledgements**

The authors would like to thank the French National Research Agency (ANR) for its support within the frame of
the SPOP project (http://spop.cirad.fr/) in Agrobiosphere program; ANR-11-AGRO-0007.

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

695        rapport. Markbrug nr. 104. Danmarks Jordbrugsforskning, Ministe-riet for Fødevarer, Landbrug og Økonomi, Tjele, Denmark.

Willigen, P. de, 2000. An analysis of the calculation of leaching and denitrification losses as practised in the NUTMON approach. Rapp.-Plant Res. Int.



**Table 1.** Main input/output variables and processes modelled in the 11 comprehensive models.

| | | WANULCAS | SNOOP | Schmidt | APSIM | Banabas | NUTMON | Ecoinvent | Meier | Brockmann | Mosier | IPCC |
|---|---|---|---|---|---|---|---|---|---|---|---|---|
| **Environmental and crop factors** | Soil and climate | | | | | | | | | | | |
| | Crop (e.g. type, root depth) | | | | | | | | | | | |
| **Non N cycling processes** | Carbon cycle | | | | | | | | | | | |
| | Plant growth | | | | | | | | | | | |
| | Water cycle | | | | | | | | | | | |
| **N Inputs** — Mineral fertiliser | Amount | | | | | | | | | | | |
| | Type (e.g. urea) | | | | | | | | | | | |
| | Application date | | | | | | | | | | | |
| | Splitting | | | | | | | | | | | |
| | Placement | | | | | | | | | | | |
| **N Inputs** — Organic fertiliser | Amount | | | | | | | | | | | |
| | Type (e.g. C/N) | | | | | | | | | | | |
| | Application date | | | | | | | | | | | |
| | Splitting | | | | | | | | | | | |
| | Placement | | | | | | | | | | | |
| | Legume N fixation | | | | | | | | | | | |
| | Atmospheric depositions | | | | | | | | | | | |
| | Non legume N fixation | | | | | | | | | | | |
| **N internal fluxes** — Residues decomposition | Previous palm -> Soil | | | | | | | | | | | |
| | Fronds -> Soil | | | | | | | | | | | |
| | Dead roots -> Soil | | | | | | | | | | | |
| | Male inflo, frond bases -> Soil | | | | | | | | | | | |
| | Legume residues -> Soil | | | | | | | | | | | |
| **N internal fluxes** — Uptake | Soil -> Oil palm | | | | | | | | | | | |
| | Soil -> Legume | | | | | | | | | | | |
| | Mineralisation / Immobilisation | | | | | | | | | | | |
| | Nitrification / Denitrification | | | | | | | | | | | |
| | Ammonification | | | | | | | | | | | |
| **N losses estimated** | Leaching | | | a | | | b | | | d | f | h |
| | N2O emissions | | | | | | b | c | | | | |
| | NH3 from fertiliser | | | | | | b | c | e | | g | |
| | Runoff | | | a | | | b | | | d | f | h |
| | NOx emissions | | | | | | b | c | | | g | |
| | N2 emissions | | | | | | b | | | | | |
| | Erosion | | | a | | | b | | | d | | |
| | NH3 from leaves | | | | | | b | | e | | | |

Legend:
- Used as input variable
- Calculated by the model
- Neither used nor calculated
- Losses with the same letter are calculated jointly





**Table 2**. Synthesis of the challenges identified in modelling the N cycle in oil palm plantations. BNF: biological N fixation.

| Challenges | Recommendations for modellers | Data available and lacking |
|---|---|---|
| **To better understand and model the N cycle during the immature period** | - To better model the magnitude and the timing of the peak of emissions<br>- To better understand and model the dynamics of N release from the residues, and the dynamics of legume N fixation, uptake, and release | - Measurements of kinetics are available for residue decomposition (Pardon et al., 2016)<br>- Knowledge is lacking concerning fluxes of N between legumes and soil, and actual losses over this period (Pardon et al., 2016) |
| **To better model the main losses through leaching, runoff, NH$_3$ volatilisation, and N$_2$O emissions** | **Leaching and runoff:**<br>- To favour a modelling approach using soil layers to obtain more precise estimates<br>- To favour a daily step approach to model the influence of timing and splitting of fertiliser application<br>- To focus on the most influential variables: soil clay content, oil palm rooting depth and oil palm N uptake<br>**NH$_3$ volatilisation:**<br>- To select emission factors more relevant to tropical conditions and perennial crops | **N$_2$O emissions:** data is still lacking for tropical conditions (Pardon et al, 2016) to allow evaluation of the models |
| **To model most of the N fluxes in order to complete the N cycle** | - **For input fluxes:** include atmospheric N deposition and BNF<br>- **For internal fluxes:** include felled palms from the previous cycle, and all the palm residues (fronds, inflorescences, roots)<br>- **For losses:** to model erosion without requiring too much data, to consider NH$_3$ emissions from leaves, to model NO$_x$ and N$_2$ even with simple models already available | - Measurements of quantities and kinetics of decomposition are already available for internal fluxes (Pardon et al., 2016).<br>- Measurements under oil palm are lacking for NO$_x$ and N$_2$ (Pardon et al, 2016) |
| **To favour ways of modelling adapted to oil palm specificities and to the objectives of the modelling** | - To favour models accounting for the whole cycle<br>- To favour a daily step approach and to integrate the spatial heterogeneity, in order to account better for the influence of fertiliser management<br>- To favour low data requirement models so they can be run easily<br>- To estimate separately the losses via each pathway to calculate its impact and to identify potential mitigation practices | |






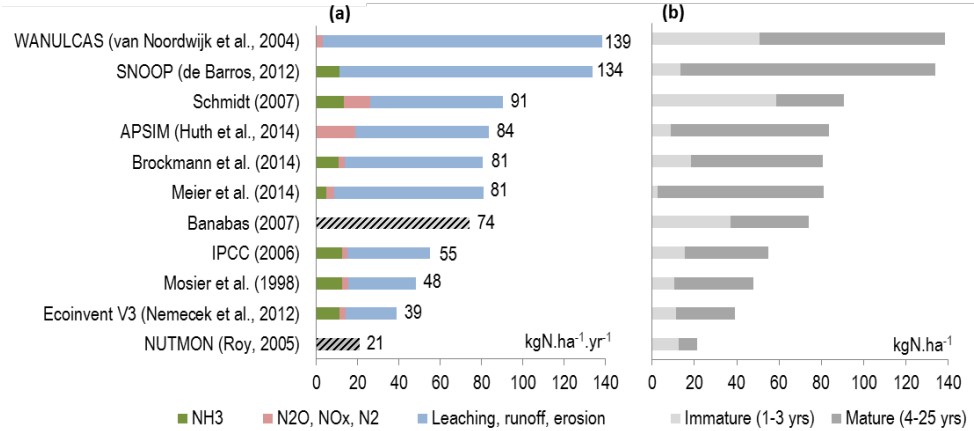

Figure 1. **Estimates of N losses by 11 models. (a) Distribution of the annual average losses between the 3 pathways**: leaching and runoff; $NH_3$ volatilisation; $N_2O$, $NO_x$, $N_2$ emissions. Overall losses of N were very variable, with an average of 77 kg N.ha$^{-1}$.yr$^{-1}$, ranging from 21 to 139 kg N.ha$^{-1}$.yr$^{-1}$. The leaching and runoff pathway was the most important of the three, corresponding to about 80% of the losses. The hatched bars represent calculations including several pathways at once: Banabas estimated jointly the 3 pathways, NUTMON estimated jointly all gaseous emissions and leaching losses were negative. SNOOP estimated $N_2$, $N_2O$, and $NO_x$ emissions as null, and APSIM and WANULCAS did not model the $NH_3$ volatilisation. **(b) Distribution of the annual average losses between the immature and the mature phases**, corresponding to 1-3 years, and 4-25 years after planting; respectively. On average, 31% of the losses occurred during the immature period, which represents 12% of the cycle duration.



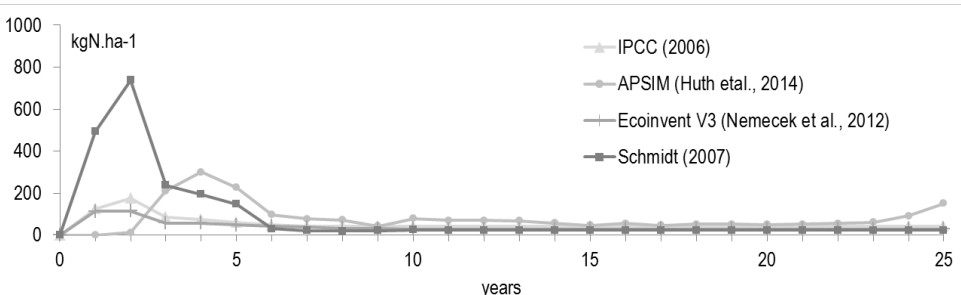


**Figure 2. Temporal patterns of N losses along the growth cycle** for 4 approaches selected to illustrate the variability of the results. Most of the models simulated maximum losses near the beginning of the cycle. The timing of the peak depended on the model, occurring between the first and the fourth year. The magnitude of the peak was very variable, up to 738 $kgN.ha^{-1}.yr^{-1}$ for Schmidt.






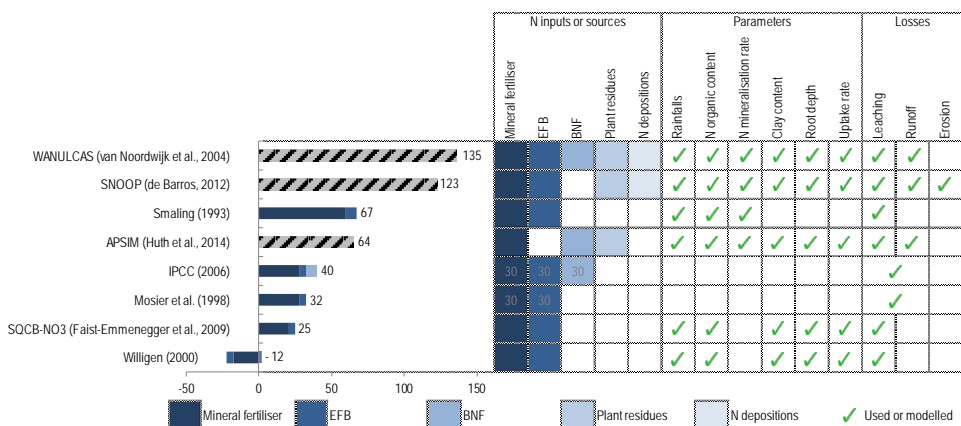

**Figure 3**. **Comparison of annual average losses through leaching and runoff, estimated by 8 sub-models**.
The average loss estimate was 59 kg N.ha$^{-1}$.yr$^{-1}$. The results represented mostly losses through leaching due to
low values for runoff losses (< 0.06 kg N.ha$^{-1}$.yr$^{-1}$). The hatched bars represent calculations which include
several sources at once: in WANULCAS, SNOOP and APSIM, all sources are considered in the same
calculation. The table shows the N inputs and parameters used by the sub-models, and emission factors for linear
relationships. Emission factors are in %, e.g. in IPCC 2006, leaching and runoff are 30% of mineral N applied.
BNF: biological N fixation; EFB: empty fruit bunches, i.e. organic fertiliser.





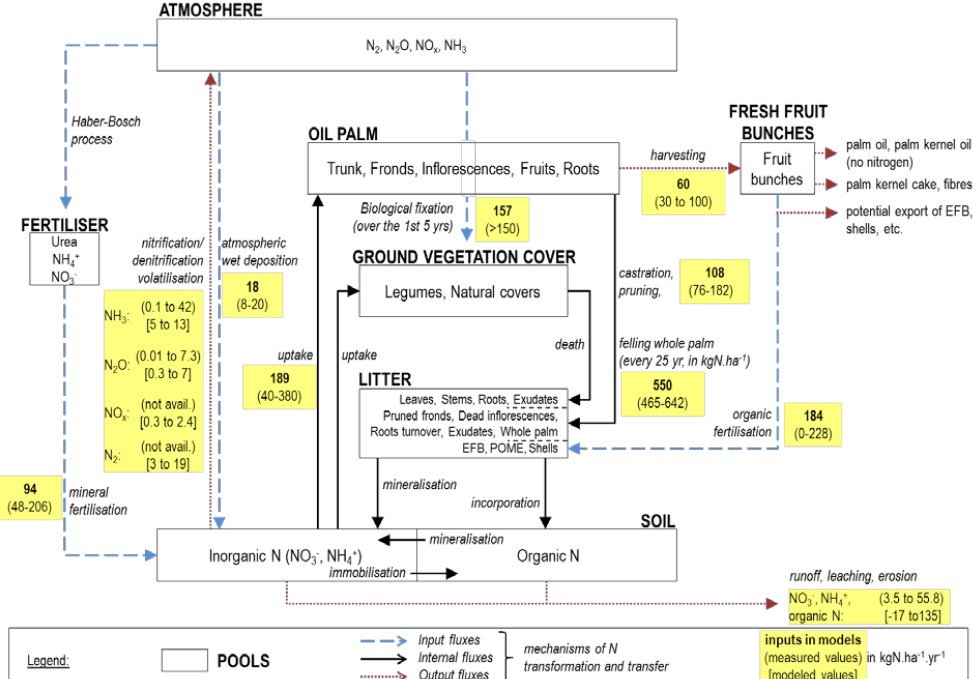

**Figure 4. Comparison of measured and modelled N losses in oil palm plantations**. The range of modelled values for leaching and runoff was wider than the one of measured values of leaching, runoff and erosion. Modelled $NH_3$ volatilisation seemed underestimated, however the maximum value of 42 kgN.ha$^{-1}$.yr$^{-1}$ was measured for mineral fertiliser applications of solely urea, while the rate of urea in our scenario was of 25% of mineral fertiliser. Modelled $N_2O$ emissions were similar to field measurements, although the minimum value was not as low. The pools are represented by the rectangles and the main fluxes are represented by the arrows. Flux values are ranges given in kgN.ha$^{-1}$.yr$^{-1}$. Measured values are from Pardon et al. (2016). POME: palm oil mill effluent; EFB: empty fruit bunches.




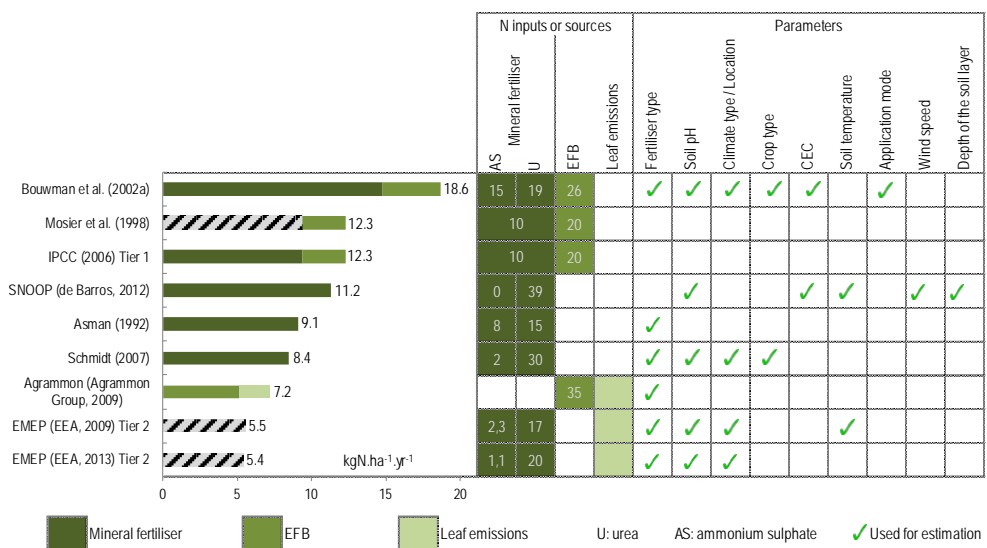

**Figure 5. Comparison of annual average losses through NH₃ volatilisation, estimated by 9 sub-models.** The average emissions from mineral fertiliser were estimated at 9.2 kgN.ha⁻¹.yr⁻¹. The emission factors for urea and ammonium sulphate differed considerably between models, ranging from 10 to 39% and 1.1 to 15%, respectively. The hatched bars represent calculations which include several sources at once: in Mosier, NH₃ emissions from mineral fertiliser include $NO_x$ emissions, and in EMEP 2009 and EMEP 2013, emissions from mineral fertiliser include those from leaves. The table shows the N inputs and parameters used by the sub-models, and emission factors for linear relationships. Emission factors are in % of N inputs. EFB: empty fruit bunches, i.e. organic fertiliser.



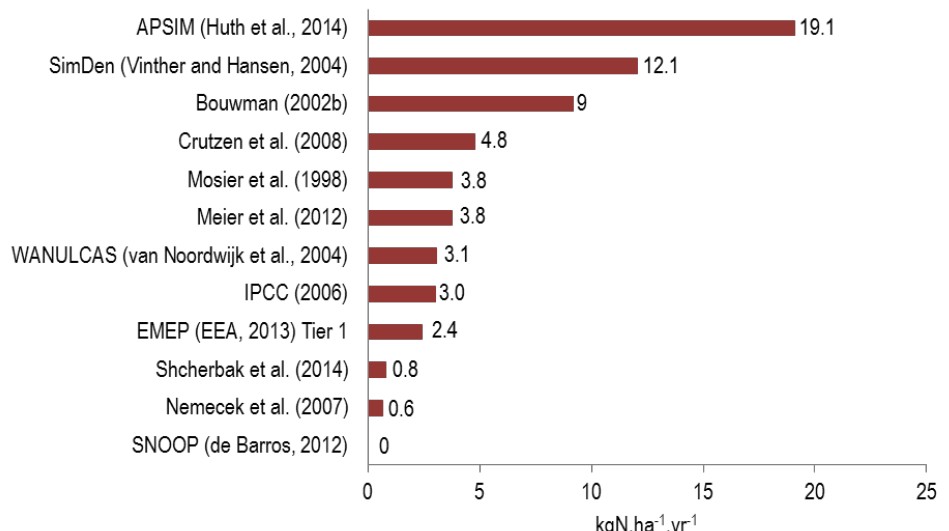

**Figure 6**. **Comparison of annual average losses through $N_2O$, $N_2$, $NO_x$ emissions, estimated by 12 sub-**
**models.** The average estimate of combined $N_2$, $N_2O$ and $NO_x$ emissions was 5.2 kgN.ha$^{-1}$.yr$^{-1}$. The wide range of
0 to 19.1 kgN.ha$^{-1}$.yr.$^{-1}$ could be explained partly because some sub-models estimated only $N_2O$ or $NO_x$, while
others calculated two or three of these gases jointly.





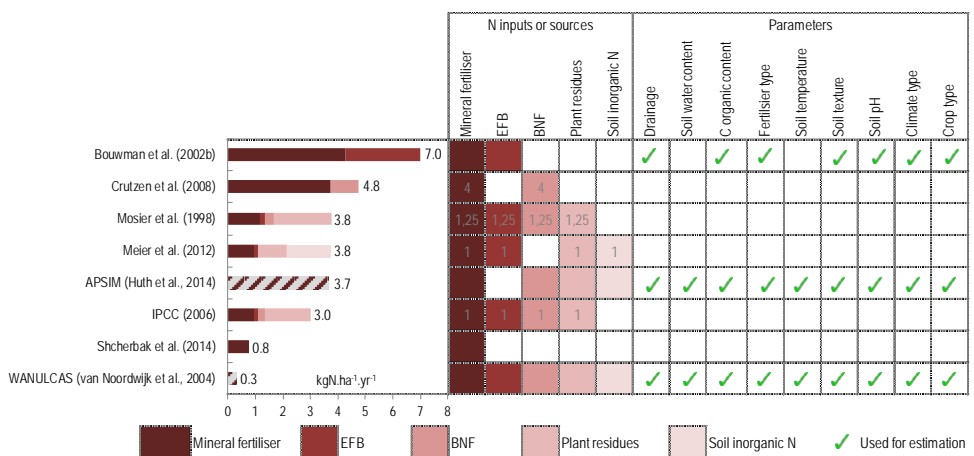

**Figure 7. Comparison of annual average losses through N$_2$O emissions, estimated by 8 sub-models.** The
average estimate was 3.4 kgN.ha$^{-1}$.yr$^{-1}$, ranging from 0.3 to 7 kgN.ha$^{-1}$.yr.$^{-1}$. For APSIM, all sources are
considered in one calculation. The table shows the N inputs and parameters used by the sub-models, and
emission factors for linear relationships. Emission factors are in % of N inputs. BNF: biological N fixation; EFB:
empty fruit bunches, i.e. organic fertiliser.





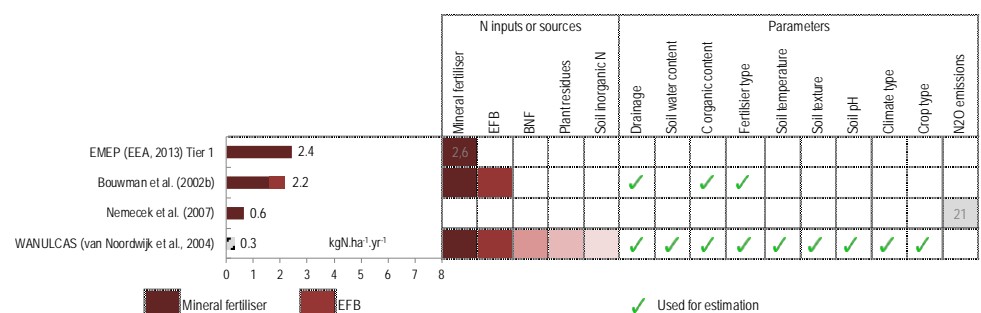

**Figure 8. Comparison of annual average losses through NO$_x$ emissions, estimated by 4 sub-models sub-models.** The average estimate was 1.4 kgN.ha$^{-1}$.yr.$^{-1}$, ranging from 0.3 to 2.4 kgN.ha$^{-1}$.yr.$^{-1}$. For Nemecek 2007, all sources are considered in one calculation. The table shows the N inputs and parameters used by the sub-models, and emission factors for linear relationships. Emission factors are in % of N inputs. EFB: empty fruit bunches, i.e. organic fertiliser.



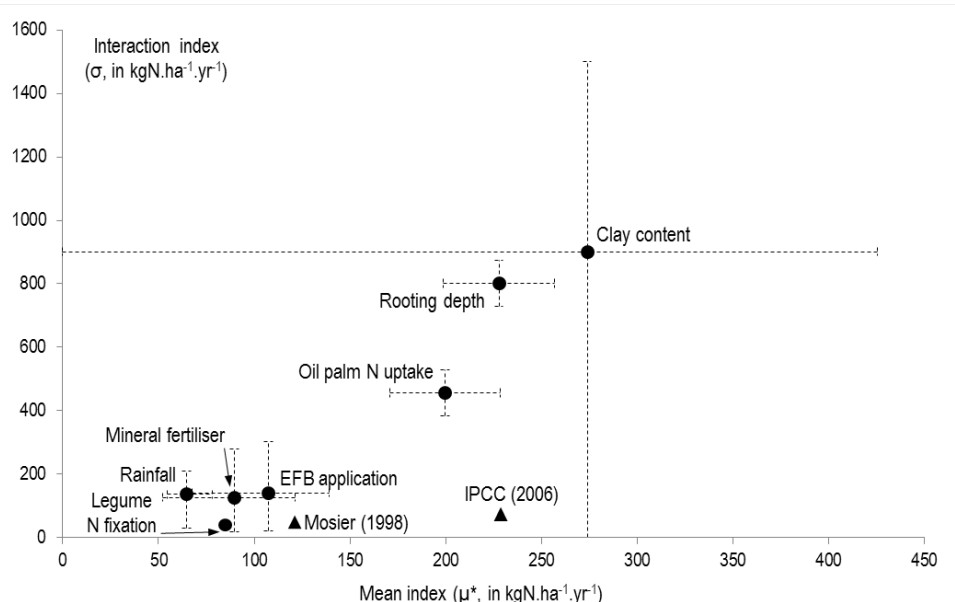

**Figure 9. Morris's sensitivity indices for 5 sub-models calculating leaching and runoff losses**. Clay content,
rooting depth, oil palm N uptake had high interaction indices, and they had the most important mean indices with
IPCC 2006 emission factor. Sub-models tested: IPCC 2006, Mosier, Smaling, Willigen and SQCB-NO3. Indices
lower than 50 kgN.ha$^{-1}$.yr$^{-1}$ are not represented. Triangles: emission factors; circles: N inputs, oil palm and
environment characteristics. EFB: empty fruit bunches, i.e. organic fertiliser.






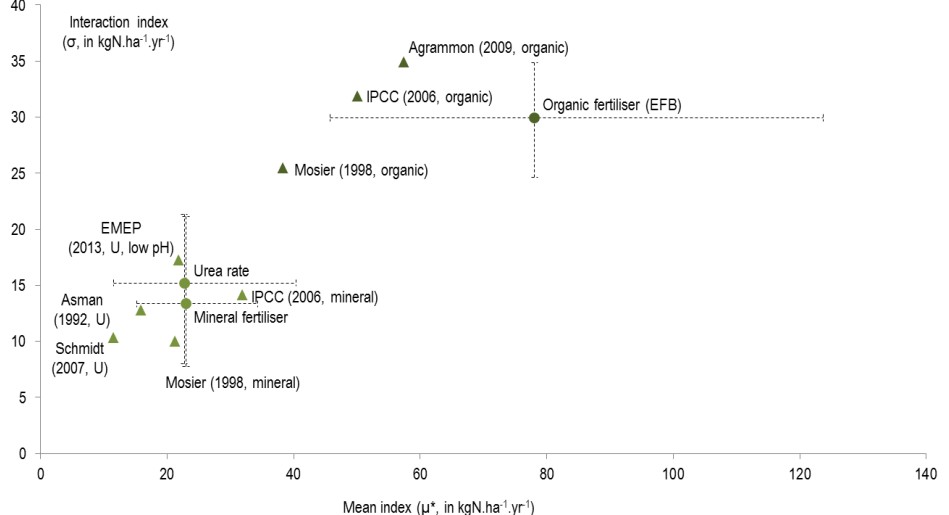

**Figure 10. Morris's sensitivity indices for sub-models calculating NH$_3$ volatilisation**. The input variables related with organic inputs (dark green) had higher Morris indices than mineral inputs (clear green). Sub-models tested: IPCC 2006, Mosier, Asman, Schmidt, Agrammon, EMEP 2009 and EMEP 2013. Indices lower than 10 kgN.ha$^{-1}$.yr$^{-1}$ are not represented. Triangles: emission factors; circles: N inputs. AS: Ammonium Sulfate; U: Urea; EFB: empty fruit bunches, i.e. organic fertiliser.



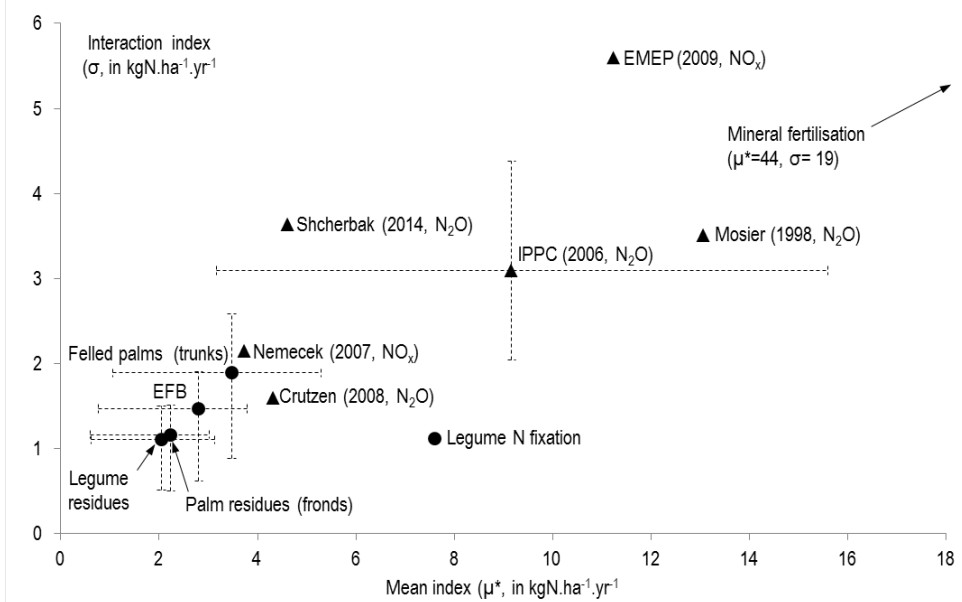

**Figure 11. Morris's sensitivity indices for sub-models calculating N₂O, NOₓ, and N₂ emissions**. Mineral

fertiliser application had the highest indices (out of this graph). For other input variables, emission factors (triangles) had higher Morris indices than N inputs (circles). Sub-models tested: Mosier, IPCC 2006, Crutzen, Meier 2014, EMEP 2013, Nemecek 2012. Indices lower than 2 kgN.ha$^{-1}$.yr$^{-1}$ are not represented. EFB: empty fruit bunches, i.e. organic fertiliser.



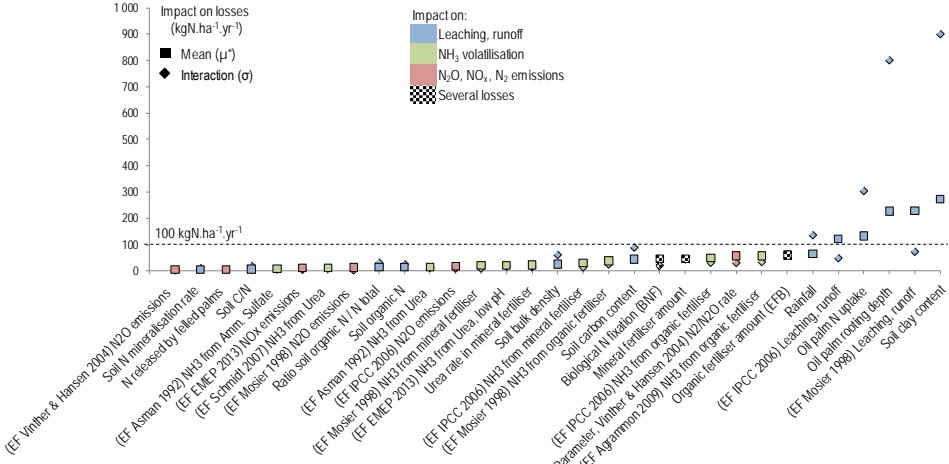

**Figure 12. Average Morris indices for 31 variables of the 19 sub-models.** The five variables having the highest influence ($\mu* > 100$ kgN.ha$^{-1}$.yr$^{-1}$) were related with leaching and runoff losses. Variables were ranked by increasing mean sensitivity index ($\mu*$). The mean effect ($\mu*$, squares) was an estimation of the linear influence of the variable on losses. The interaction effect ($\sigma$, diamonds) was an estimation of non-linear and/or interaction effect(s) of the variable on losses. Variables with $\mu* < 5$ kgN.ha$^{-1}$.yr$^{-1}$, i.e. 16 variables, are not represented. EF: emission factor; BNF: biological N fixation; EFB: empty fruit bunches, i.e. organic fertiliser.