# Peer review of "Quantifying nitrogen losses in oil palm plantations: models and challenges"

_Biogeosciences, 2016_

## Referee Comment (RC1) · M. van Noordwijk (Referee) · 24 Jun 2016

Comments on bg-2016-177: Quantifying nitrogen losses in oil palm plantations: models and challenges by L. Pardon et al.

With the rapid expansion of oil palm the environmental consequences beyond initial conversion need to be quantified. The manuscript describes a comprehensive comparison of existing models that can be used to predict N losses from oil palm, both at qualitative and quantitative level. The tables of which processes are included (or not) can help in further model development, borrowing ideas within the modelling community, retaining what seems to work best. As there is no established empirical data set that can serve as comparison, it is not easy to see which models over- or underestimate the overall loss, beyond what the discussion indicates. The manuscript will

hopefully trigger further critical analysis, and is worth publication in Biogeosciences. The manuscript takes the various models at 'face value', without comparing the underlying equations. As leaching is identified as the dominant process of N loss across the models, it will be good to provide further detail on why the various models provide different estimates. I assume that the basic equation for leaching losses as the net vertical water transport multiplied by average concentration of mineral N in soil solution holds true across all models. Variation in results could then be based on differences in ET (with vertical water transport equal to rainfall -ET) or concentration. The latter will vary with N form (fraction of mineal N as ammonium vs nitrate), sorption of ammonium and nitrate to the soil, or any preferential flow that allows water to bypass mineral N. Please add a paragraph or two to the dicussion that takes this debate a bit further. Identification oc clay content as key variable points to interactions via sorption and/or microbial dynamics – again, a bitfurther discussion of the underlying mechanisms that are included in the model is of interest. One would expect pH to play a substantial role, as it influences nitrification as a process – is that included in any of the models? It may be good to include some further words of caution to use of the 'portfolio of models' rather than a single one – in the absence of further empirical studies. There is no specific reason to believe that the average is closer to the truth than any of the values in the range of results obtained...

Overall I recommend the manyscript to be published with some minor revisions.

Comments that may be taken into account in a final version: Around line 45 it may be good to give some a priori reasons why porcess models adjusted to temprate conditions might not work, without adjustement, to tropical conditions: faster decomposition, different dominant clay minerology, different soil microflora and fauna, ... reference could be made to: Richards, M., Metzel, R., Chirinda, N., Ly, P., Nyamadzawo, G., Vu, Q.D., de Neergaard, A., Oelofse, M., Wollenberg, E., Keller, E. and Malin, D., 2016. Limits of agricultural greenhouse gas calculators to predict soil N2O and CH4 fluxes in tropical agriculture. Scientific reports, 6.

Around line 48 a distinction may be made between models that are primarily used for 'response to management' type studies, and those that are supposed to provide an accurate mean

Line 74 reference for Morris please

Line 240 Please clarify which reason applied. Aren't all models 'open source'? to which model

Line 245 a range from 10 to 190% of initial estimate may get one into pretty extreme conditions on the low side, and pretty mild one on the high side, as many effects are multiplicative rather than additive... This may require some text in the discussion.

Line 267 Please provide the calculated yield levels and average annual external N inputs that correspond with this result, along with rainfall and ET

Line 336 Was application of EFB only considered at planting time?

Line 375 Any differentiation by clay minerology in any of the models?

Line 446 If fertilizer is added close to the trunk in a zone with average of above-average water infiltration, a high leaching loss is to be expected, unless there are contravening processes of preferential water uptake in this zone...

Line 490 The Richards reference cited above suggests that there are real knowledge gaps on the N2O emissions for any current model

Typo's Line 429 Some words missing?

---

## Referee Comment (RC2) · H. Corley (Referee) · 29 Jun 2016

General comments

This is a very useful paper. Nitrogen losses have a large economic cost, in addition to any environmental aspects, so the subject should be relevant to all concerned with the crop. I think they could be more critical in the comparison of models, discarding those which clearly do not give useful output. From my reading of the results, the Schmidt model is clearly the best, and I think they could identify that as the basis for further model development.

Specific comments

Line 30 - The comparison with field measurements is the only useful test of the model's

validity.

Section 3.3 - Sensitivity analysis: Combined results for all models, as set out here, seem rather irrelevant. Given that some of the models give poor results in comparison with actual measurements, it would be better to first identify models which might be useful (Banabas and Schmidt look best - see line 452), and then assess the sensitivity of those models only.

Lines 495-500 - I agree, and thus conclude that the Schmidt model is better than Banabas. However, Schmidt still combines leaching, run-off and erosion, so could be improved.

Line 562 - It is probably correct that leaching is the main loss pathway, and also probably correct that losses are particularly high during the immature stage, but I do not believe there are sufficient published data to support quite such a definite statement as here.

Line 563 - Losses from legume cover crop presumably only become significant when the crop starts to be shaded out and to decay.

Figure 2 - Only the pattern from the Schmidt model is similar to that shown by Pardon et al. (2016).

Figure 12 - a logarithmic scale on the Y-axis would allow differences between the smaller values to be seen

Technical comments and corrections

Line 39 - Corley & Tinker should be 2003, not 2008

Line 58 - I think this should read "...the delay between inflorescence meristem initiation..."

Figure 1, etc - These figures might be improved by including an additional line showing the range of published data found by Pardon et al. (2016).
The emission factors in the figures are almost illegible. Greater contrast between the text and the coloured boxes is needed.

---

## Short Comment (SC1) · 28 Jul 2016

Referee: With the rapid expansion of oil palm the environmental consequences beyond initial conversion need to be quantified. The manuscript describes a comprehensive comparison of existing models that can be used to predict N losses from oil palm, both at qualitative and quantitative level. The tables of which processes are included (or not) can help in further model development, borrowing ideas within the modelling community, retaining what seems to work best. As there is no established empirical data set that can serve as comparison, it is not easy to see which models over- or under-estimate the overall loss, beyond what the discussion indicates. The manuscript will hopefully trigger further critical analysis, and is worth publication in Biogeosciences.

Reply: Thank you for your constructive and precise comments which improved the

clarity of the manuscript. They also invited us to emphasise some issues which can be useful then to target potential improvements in modelling N losses in oil palm plantations.

Referee: The manuscript takes the various models at 'face value', without comparing the underlying equations. As leaching is identified as the dominant process of N loss across the models, it will be good to provide further detail on why the various models provide different estimates. I assume that the basic equation for leaching losses as the net vertical water transport multiplied by average concentration of mineral N in soil solution holds true across all models. Variation in results could then be based on differences in ET (with vertical water transport equal to rainfall -ET) or concentration. The latter will vary with N form (fraction of mineal N as ammonium vs nitrate), sorption of ammonium and nitrate to the soil, or any preferential flow that allows water to bypass mineral N. Please add a paragraph or two to the dicussion that takes this debate a bit further.

Reply: Thank you for this comment, we agree that it is useful to discuss more about the reasons why leaching estimates are different, since it is the dominant N loss pathway. The different ways of modelling leaching by the models are described in the section 2.1.2. in Material and method (lines 139-152), and only the three process-based models (APSIM, SNOOP, and WANULCAS) simulate explicitly water flows. Therefore we added these precisions (line 454) to focus on two processes likely to drive these differences in leaching estimates: "For instance, interactions between zoning and N inputs might occur in WANULCAS, as the mineral N input from fertiliser was applied close to the trunks where water infiltration might be higher due to the stemflow. Another potential important interaction might happen with the simulation of N immobilisation and mineralisation in soil. Indeed, in WANULCAS, mineralisation of residues and empty fruit bunches caused high losses through leaching in the first years of the cycle, while in APSIM, immobilisation of N dominated the dynamics over several years and losses through leaching were delayed and much lower."

Referee: Identification oc clay content as key variable points to interactions via sorption and/or microbial dynamics – again, a bitfurther discussion of the underlying mechanisms that are included in the model is of interest. One would expect pH to play a substantial role, as it influences nitrification as a process – is that included in any of the models?

Reply: Thank you for this comment. As the sub-models tested in the sensitivity analysis are not process-based models, we could not discuss the underlying mechanisms. However, we are preparing another study focused on a sensitivity analysis of APSIM-Oil palm in which we tested variables and discussed the underlying processes simulated. Therefore, we added these 3 sentences in the discussion (Line 534): "The sub-models included in the sensitivity analysis were regression models which did not simulate processes explicitly. Some parameters not taken into account in these models may have an influence in other process-based models. Therefore, it could be interesting to perform complementary sensitivity analyses focused on process-based models, such as APSIM."

Referee: It may be good to include some further words of caution to use of the 'portfolio of models' rather than a single one – in the absence of further empirical studies.

Reply: Thank you for this proposal. We added this sentence in the discussion (L 468): "Therefore, our results call for caution with regard to the choice of a single model to simulate N losses in oil palm. In absence of further empirical studies available to test these models, we would recommend to use several models to perform N losses predictions."

Referee: There is no specific reason to believe that the average is closer to the truth than any of the values in the range of results obtained...

Reply: Yes, we agree with your comment. However, we identified the value of 74 kg N.ha-1.yr-1 as a plausible estimate, following a nitrogen budget reasoning (assuming no nitrogen is stored in the soil over the growth cycle, L 430-438), rather than because

it would be the average of the results for all models.

Referee: Overall I recommend the manyscript to be published with some minor revisions. Comments that may be taken into account in a final version: Around line 45 it may be good to give some a priori reasons why porcess models adjusted to temprate conditions might not work, without adjustement, to tropical conditions: faster decomposition, different dominant clay minerology, different soil microflora and fauna, ... reference could be made to: Richards, M., Metzel, R., Chirinda, N., Ly, P., Nyamadzawo, G., Vu, Q.D., de Neergaard, A., Oelofse, M., Wollenberg, E., Keller, E. and Malin, D., 2016. Limits of agricultural greenhouse gas calculators to predict soil N2O and CH4 fluxes in tropical agriculture. Scientific reports, 6.

Reply: Thank you for this very relevant reference. We added the reference in the text (L 57), and we mentioned explicitly a priori reasons (as examples) why N dynamics are expected to be different under tropical climate and perennial crops (L 44): "N losses under perennial tropical crops are expected to follow specific dynamics, given for instance the highest temperature and rainfall, and the high amount of crop residues recycled over the growth cycle."

Referee: Around line 48 a distinction may be made between models that are primarily used for 'response to management' type studies, and those that are supposed to provide an accurate mean

Reply: Ok, we completed a sentence to add this interesting distinction (L 50): "Nowadays, models are also widely used to estimate the emission of pollutants for the purpose of environmental assessment, aiming either at simulating accurate mean emissions, or at estimating the impact of management practices on emissions."

Referee: Line 74 reference for Morris please

Reply: Reference added, thank you.

Referee: Line 240 Please clarify which reason applied. Aren't all models 'open

source'? to which model

Reply: Ok, we clarified the sentence (L 245): "Process-based models were not included in the sensitivity analysis as the source code of SNOOP was not accessible and APSIM and WANULCAS were not directly programmable without adapting the models structure to run the sensitivity analysis, which fell beyond the scope of this study."

Referee: Line 245 a range from 10 to 190

Reply: Ok, we added 2 sentences in the discussion to raise awareness on these wide ranges and remind the reason of choosing such ranges in the sensitivity analysis (L531-534) : "When the sub-models did not specify ranges for emission factors, we used ranges from -90

Referee: Line 267 Please provide the calculated yield levels and average annual external N inputs that correspond with this result, along with rainfall and ET

Reply: Ok, we added the ranges for these values (L274). "Annual estimates were 20-25 t of fresh fruit bunches.ha-1.yr-1 for yield, 132-147 kg N.ha-1.yr-1 of N inputs (mineral fertiliser, atmospheric deposition, biological N fixation, empty fruit bunches and previous felled palms), 2407 mm.yr-1 of rainfall and 932-1545 mm.yr-1 of evapotranspiration."

Referee: Line 336 Was application of EFB only considered at planting time?

Reply: Yes, as we said at line 211 in MM, "Organic fertiliser, i.e. empty fruit bunches, was applied around the palms for the first two years as a typically used rate of 184 kg N.ha-1.yr-1."

Referee: Line 375 Any differentiation by clay minerology in any of the models?

Reply: There is no differentiation by clay mineralogy in any of the models.

Referee: Line 446 If fertilizer is added close to the trunk in a zone with average of above-average water infiltration, a high leaching loss is to be expected, unless there

are contravening processes of preferential water uptake in this zone...

Reply: Thank you, we added this more precise description in the discussion (as cited above, L454): "For instance, interactions between zoning and N inputs can occur in WANULCAS, as the N inputs from fertiliser is applied close to the trunks where water infiltration might be higher due to the stemflow."

Referee: Line 490 The Richards reference cited above suggests that there are real knowledge gaps on the N2O emissions for any current model

Reply: Our sentence was more related to the need of modelling N2O, NOx and N2 emissions in a comprehensive way. But you are right, in order to contextualise this discussion, we added a sentence and a reference about N2O modelling (L505): "Finally, despite the difficulties of understanding and simulating the complexity of processes driving N2O emissions (Butterbach-Bahl et al., 2013), N2O, NOx and N2 should be modelled in a more comprehensive and systematic way."

Referee: Typo's Line 429 Some words missing?

Reply: Thank you, we checked the Line 429, but it seemed correct to us.

See final comment for the revised manuscript containing all the revisions.

---

## Short Comment (SC2) · 28 Jul 2016

Referee: General comments This is a very useful paper. Nitrogen losses have a large economic cost, in addition to any environmental aspects, so the subject should be relevant to all concerned with the crop.

Reply: Thank you very much for your comments. They notably invited us to step back on our results and discussion, and they improved the clarity of the figures.

Referee: I think they could be more critical in the comparison of models, discarding those which clearly do not give useful output. From my reading of the results, the Schmidt model is clearly the best, and I think they could identify that as the basis for further model development.

[Figure]

Reply: We understand that clearer recommendations about models to be used or not could add a value to this study. However, on the contrary, our other referee proposed a cautious approach regarding recommendations: "It may be good to include some further words of caution to use of the 'portfolio of models' rather than a single one – in the absence of further empirical studies." Indeed, we could not have access to a comprehensive dataset of measured N losses in one specific site. Therefore, we rather chose to keep a cautious approach, adding this sentence in the discussion (L 468): "Therefore, our results call for caution with regard to the choice of a single model to simulate N losses in oil palm. In absence of further empirical studies available to test these models, we would recommend to use several models to perform N losses predictions." Please, see below for specific comments and replies about Schmidt model.

Referee: Specific comments Line 30 - The comparison with field measurements is the only useful test of the model's validity.

Reply: Yes, we agree. This led us to favour a cautious approach in the discussion concerning a potential ranking of models, because the only available measurements data to assess the models were overall ranges mentioned in Pardon et al. (2016), which summarised a great variety of conditions.

Referee: Section 3.3 - Sensitivity analysis: Combined results for all models, as set out here, seem rather irrelevant. Given that some of the models give poor results in comparison with actual measurements, it would be better to first identify models which might be useful (Banabas and Schmidt look best - see line 452), and then assess the sensitivity of those models only.

Reply: Thank you, we understand your advice. However, two points led us to consider still all models in the sensitivity analysis: first, it provides a more comprehensive comparison of the models throughout the paper; second, as the sensitivity analysis is done at the level of the sub-models (each one being potentially used by several comprehensive models), the link between performances of comprehensive models and performances of their sub-models is not always obvious, which means that the selection of the sub-models for the sensitivity analysis would not be straightforward and subject to discussion.

Referee: Lines 495-500 - I agree, and thus conclude that the Schmidt model is better than Banabas. However, Schmidt still combines leaching, run-off and erosion, so could be improved.

Reply: Ok, thank you for your suggestion. In order to be more explicit, we hence added this sentence at the end of the paragraph (L518): "Regarding these criteria, Schmidt model is the most comprehensive and detailed model, but could be improved by modelling separately losses through erosion, runoff and leaching."

Referee: Line 562 - It is probably correct that leaching is the main loss pathway, and also probably correct that losses are particularly high during the immature stage, but I do not believe there are sufficient published data to support quite such a definite statement as here.

Reply: Yes, we agree, and we moderated the statement replacing "is particularly high" by "is likely to be high".

Referee: Line 563 - Losses from legume cover crop presumably only become significant when the crop starts to be shaded out and to decay.

Reply: Yes, we agree, and we replaced "immature phase" by "young phase" in order to refer to a longer period.

Referee: Figure 2 - Only the pattern from the Schmidt model is similar to that shown by Pardon et al. (2016).

Reply: Yes, however we cannot really use the pattern shown in Pardon et al. (2016) to assess these models as it is in itself another model. Moreover, the similarity mainly comes from the fact that the pattern in Pardon et al. (2016) is modelled using a N budget approach accounting for most of the N fluxes, which is the same approach than

Schmidt.

Referee: Figure 12 - a logarithmic scale on the Y-axis would allow differences between the smaller values to be seen

Reply: Thank you for the idea, we changed the Y-axis to a logarithmic scale, which improves greatly the clarity of the figure.

Referee: Technical comments and corrections Line 39 - Corley  Tinker should be 2003, not 2008

Reply: Thank you, we corrected the date.

Referee: Line 58 - I think this should read "...the delay between inflorescence meristem initiation..."

Reply: Yes, we agree, and we modified the sentence.

Referee: Figure 1, etc - These figures might be improved by including an additional line showing the range of published data found by Pardon et al. (2016).

Reply: Thank you for the idea, we added the ranges of measured data to Figure 3, 5 and 7. We did not add ranges of measured data to Figure 1 to avoid misinterpretations, as the measured values do not include N2 whereas modelled values do.

Referee: Figure 1, etc - The emission factors in the figures are almost illegible. Greater contrast between the text and the coloured boxes is needed

Reply: Ok, we increased the contrasts.

Please, see the last comment for the revised version of the manuscript.

---

## Short Comment (SC3) · 28 Jul 2016

We thank our referees for their precise and constructive comments. We revised our manuscript taking into account almost all comments. The comments and advices really helped to make the manuscript clearer, to depeen the main issues to be discussed, and to improve the figures.

Please, see the point-by-point replies for all details of the discussion, and consult the revised version attached to this comment.

Please also note the supplement to this comment:
http://www.biogeosciences-discuss.net/bg-2016-177/bg-2016-177-SC3-supplement.pdf

[Figure]

[Figure]

**Supplement:**

**Quantifying nitrogen losses in oil palm plantations: models and challenges**

**Authors: First names Last names**

**Lénaïc Pardon[1], Cécile Bessou[1], Nathalie Saint-Geours[2], Benoît Gabrielle[3], Ni'matul Khasanah[4], Jean-Pierre Caliman[1,5], Paul N. Nelson [6]**

[1]CIRAD, UPR Systèmes de pérennes, F-34398 Montpellier, France
emails : lenaic.pardon@cirad.fr; cecile.bessou@cirad.fr

[2]ITK, CEEI CAP ALPHA Avenue de l'Europe, 34830 Clapiers, France
email : nathalie.saint-geours@itk.fr

[3]AgroParisTech, INRA, UMR EcoSys, 78850 Thiverval-Grignon, France
email: benoit.gabrielle@agroparistech.fr

[4]World Agroforestry Centre (ICRAF), Southeast Asia Regional Programme, Bogor, Indonesia
email: j

[5]SMART Research Institute, Jl. Tengku Umar 19, Pekanbaru, 28112, Indonesia
email: j.p.caliman@sinarmas-agri.com

[6]College of Science and Engineering, James Cook University, Cairns 4878 Qld, Australia
email: paul.nelson@jcu.edu.au

*Correpondence to:* Cécile Bessou (cecile.bessou@cirad.fr)

**Key words:** Model testing, Oil palm, Nitrogen budget, Nitrogen losses

**Abstract.** Oil palm is the most rapidly expanding tropical perennial crop. Its cultivation raises environmental concerns, notably related to the use of nitrogen (N) fertilisers and associated pollution and greenhouse gas emissions. While numerous and diverse models exist to estimate N losses from agriculture, very few are available for tropical perennial crops. Moreover, there has been no critical analysis of the performances of existing models in the specific context of tropical perennial cropping systems. We assessed the capacity of 11 models and 29 sub-models to estimate N losses in a typical oil palm plantation over a 25-year-growth cycle, through leaching and runoff, and emissions of $NH_3$, $N_2$, $N_2O$, and $NO_x$. Estimates of total N losses were very variable, ranging from 21 to 139 kg $N.ha^{-1}.yr^{-1}$. On average, 31% of the losses occurred during the first three years of the cycle. Leaching comprised about 80% of the losses. Based on a comprehensive Morris sensitivity analysis, the most influential variables were soil clay content, rooting depth and oil palm N uptake. We also compared model estimates with published field measurements. Many challenges remain to model more accurately processes related to the peculiarities of perennial tropical crop systems such as oil palm.

**1. Introduction**

Oil palm is the most rapidly expanding tropical perennial crop. The area of land under oil palm, currently approximately 19 Mha, has been rising at 660,000 ha./year[-1] over the 2005-2014 period (FAOSTAT 2014) and is likely to continue rising until 2050 (Corley 2009). This increase raises significant environmental concerns.-

Beside issues related to land-use changes and the oxidation of peat soils when establishing plantations, the cultivation of oil palm can generate adverse environmental impacts, in particular through the use of nitrogen (N) fertilisers. The latter are associated with pollution risks of ground and surface waters, and emissions of greenhouse gases (Choo et al., 2011; Comte et al., 2012; Corley and Tinker, 2003̶8). An accurate estimation of N

40    losses from palm plantations is hence critical for reliable assessment of their environmental impacts. Models are necessary for such estimation, because comprehensive direct measurement of losses is prohibitively difficult and expensive.

While a number of models exist to estimate N losses from agricultural fields, they mostly pertain to temperate climate conditions and annual crops. N losses under perennial tropical crops are expected to follow specific

45    dynamics, given for instance the highest temperature and rainfall, and the high amount of crop residues recycled over the growth cycle. Yet, fF̶ew models are available for tropical crops, and even fewer for perennial tropical crops (Cannavo et al., 2008). Such models, in particular mechanistic ones, were primarily developed for research purposes, in order to simulate crop growth related to biogeochemical cycles and gain insight into the underlying processes. Nowadays, models are also widely used to estimate the emission of pollutants for the purpose of

50    environmental assessment, aiming either at simulating accurate mean emissions, or at estimating the impact of management practices on emissions. 
[revised manuscript text omitted]
. For instance, interactions between zoning and N inputs might occur in WANULCAS, as the mineral N input from fertiliser was applied close to the trunks where water infiltration might be higher due to the stemflow. Another potential important interaction might happen with the simulation of N immobilisation and mineralisation in soil. Indeed, in WANULCAS, mineralisation of residues and empty fruit bunches caused high losses through leaching in the first years of the cycle, while in APSIM, immobilisation of N dominated the dynamics over several years and losses through leaching were delayed and much lower. However, more work would be necessary to better understand how the structure of the models could lead to overestimates of leaching.

For estimates closer to the plausible 74 kg N.ha$^{-1}$.yr$^{-1}$, the results hide several cases. APSIM estimated a plausible overall loss of 84 kg N.ha$^{-1}$.yr$^{-1}$, but leaching seemed overestimated compared to measured values. This was very likely because some other fluxes were not taken into account, such as NH$_3$ volatilisation and N input through empty fruit bunches. Similarly, Meier 2014 and Brockmann had plausible overall loss estimates, but leaching losses seemed overestimated, while neither N$_2$ emissions, nor N input through biological N fixation, were taken into account. Schmidt and Banabas estimates seemed to be plausible and they accounted for most of the fluxes. Modelled N$_2$O emissions were similar to field measurements, although the minimum modelled emissions were still higher than the minimum losses measured in the field. Therefore, our results call for caution with regard to the choice of a single model to simulate N losses in oil palm. In absence of further empirical studies available to test these models, we would recommend to use several models to perform N losses predictions.

[revised manuscript text omitted]

Reducing the uncertainty of leaching modelling is an important challenge, as about 80% of the total losses came from leaching, according to the models, and results were very variable across models. Models should be better adapted to the oil palm system, as some regression models were out of their validity domain. The sensitivity analysis also showed that the most influential variables, upon which research should focus, were soil clay content, oil palm rooting depth and oil palm N uptake. When the sub-models did not specify ranges for emission factors, we used ranges from -90% to +90% of the nominal value to perform the sensitivity analysis. The upper and lower limits of these ranges corresponded hence to extreme conditions, but they allowed on the other hand an exploration of the sensitivity in a wide range of conditions. The sub-models included in the sensitivity analysis were regression models which did not simulate processes explicitly. Some parameters not taken into account in these sub-models may have an influence in other process-based models. Therefore, it could be interesting to perform complementary sensitivity analyses focused on process-based models, such as APSIM.

[revised manuscript text omitted]

---

## Author Response (AR1)

**Point-by-point reply to Associate Editor's comments (31/08/2016)**

Thanks for all your propositions of reformulation which really improved the clarity of the manuscript, the quality of the language, and made some sentences more straightforward.

We accepted all your propositions, except in one case where it could be confusing: L292, we kept "the leaching and runoff pathway was the most important of the three" instead of "leaching and runoff were the most important of the three N loss pathways", because in our study leaching and runoff belong to a single pathway (the two other pathways being "$NH_3$ volatilisation" and "$N_2O$, $NO_x$ and $N_2$ emissions").

For L483: We agree that the formulation was confusing. We changed to this sentence: "Lastly, the models that came up with a plausible estimate of overall N losses, i.e. close to 74 kg N.ha-1.yr-1, showed large differences in single N flux sizes."

For L548: We agree that this sentence was confusing, and we proposed improvements to clarify it: "Regarding these criteria, the Schmidt model appeared the most comprehensive and detailed one, as it distinguishes between six N fluxes. However, this model could be improved by modelling separately losses through erosion, runoff and leaching, i.e. calculating a total of eight N fluxes."

In Figure 12, we checked that mean and sigma panels are both in logarithmic scale.

In order to reduce the amount of text in the figures, we reduced the number of words of about 30% in the figure 4. In addition, we suppressed the full references in all the figures, as you proposed.

We also did a final check for language throughout the manuscript, with a special focus on the discussion as you advised.

We hope that these final revisions will allow this paper to be published in BG.

Kind regards,

Below follows the final version with tracked changes.

**Quantifying nitrogen losses in oil palm plantations: models and challenges**

Authors: First names Last names

**Lénaïc Pardon[1], Cécile Bessou[1], Nathalie Saint-Geours[2], Benoît Gabrielle[3], Ni'matul Khasanah[4], Jean-Pierre Caliman[1,5], Paul N. Nelson [6]**

[1]CIRAD, UPR Systèmes de pérennes, F-34398 Montpellier, France

emails : lenaic.pardon@cirad.fr; cecile.bessou@cirad.fr

[2]ITK, CEEI CAP ALPHA Avenue de l'Europe, 34830 Clapiers, France

email : nathalie.saint-geours@itk.fr

[3][1]UMR ECOSYS, INRA, AgroParisTech, Université Paris-Saclay, 78850, Thiverval-Grignon, France

email: benoit.gabrielle@agroparistech.fr

[4]World Agroforestry Centre (ICRAF), Southeast Asia Regional Programme, Bogor, Indonesia

email: j

[5]SMART Research Institute, Jl. Tengku Umar 19, Pekanbaru, 28112, Indonesia

email: j.p.caliman@sinarmas-agri.com

[6]College of Science and Engineering, James Cook University, Cairns 4878 Qld, Australia

email: paul.nelson@jcu.edu.au

*Correpondence to:* Cécile Bessou (cecile.bessou@cirad.fr)

**Key words:** Model testing, Oil palm, Nitrogen budget, Nitrogen losses

**Abstract.** Oil palm is the most rapidly expanding tropical perennial crop. Its cultivation raises environmental concerns, notably related to the use of nitrogen (N) fertilisers and the associated pollution and greenhouse gas emissions. While numerous and diverse models exist to estimate N losses from agriculture, very few are currently available for tropical perennial crops. Moreover, there is a lack of critical analysis of their performance in the specific context of tropical perennial cropping systems. We assessed the capacity of 11 models and 29 sub-models to estimate N losses in a typical oil palm plantation over a 25-year-growth cycle, through leaching and runoff, and emissions of $NH_3$, $N_2$, $N_2O$, and $NO_x$. Estimates of total N losses were very variable, ranging from 21 to 139 kg $N.ha^{-1}.yr^{-1}$. On average, 31% of the losses occurred during the first three years of the cycle. Nitrate Lleaching accounted for about 80% of the losses. A comprehensive Morris sensitivity analysis showed the most influential variables to be soil clay content, rooting depth and oil palm N uptake. We also compared model estimates with published field measurements. Many challenges remain to model more accurately processes related to the peculiarities of perennial tropical crop systems such as oil palm.

**1. Introduction**

Oil palm is the most rapidly expanding tropical perennial crop. The area of land under oil palm, currently amounting to approximately 19 Mha, has been rising at 660,000 ha./year[-1] over the 2005-2014 period (FAOSTAT 2014), and this trend is likely to continue  until 2050 (Corley 2009). This increase raises significant environmental concerns. Beside issues related to land-use changes and the oxidation of peat soils when establishing plantations, the cultivation of oil palm can generate adverse environmental impacts, in particular through the use of nitrogen (N) fertilisers. The latter are associated with pollution risks for ground and surface waters, and emissions of greenhouse gases (Choo et al., 2011; Comte et al., 2012; Corley and Tinker, 2008). As a result, an accurate estimation of N losses from palm plantations is critical to a reliable assessment of their environmental impacts. Models appear necessary in this process because comprehensive direct measurements of N losses are too difficult and resource-intensive to be generalised.

While a number of models exist to estimate N losses from agricultural fields, they mostly pertain to temperate climate conditions and annual crops. N losses under perennial tropical crops are expected to follow specific dynamics, given for instance the higher ranges of temperature and rainfall experienced in these climatic zones, and the high amount of crop residues recycled over the growth cycle. Yet, Few models are available for tropical crops, and even fewer for perennial tropical crops (Cannavo et al., 2008). Such models, in particular mechanistic ones, were primarily developed for research purposes, in order to simulate crop growth as affected by biogeochemical processes, and to gain insight into the underlying processes. Nowadays, models are also widely used to estimate the emission of pollutants for the purpose of environmental assessment, aiming either at more accurate estimates of mean emissions, or at evaluating the impact of certain management practices on emissions. Different types of models are used, ranging from highly complex process-based models to more simple operational models such as empirical regressions. Despite some consensus and recommendations regarding best practices for the modelling of field emission s, notably within the framework of life cycle assessment (e.g. IPCC, 2006; EC ILCD, 2011), there has not been any comprehensive review and comparison of potentially useful models for environmental assessment. Moreover, various publications pinpointed the need for models that are better adapted to tropical crops in the estimation of  field emissions  (Basset-Mens et al., 2010; Bessou et al., 2013; Cerutti et al., 2013; Richards et al., 2016). To improve field emissions modelling in oil palm plantations, we need to determine the potential applicability and pitfalls of state-of-the art models regarding N cycling and losses in these systems.

Most environmental impact assessment methods, such as life cycle assessment, consider perennial systems to behave similarly to annual ones. Following this assumption, the inventory data on the farming system  are generally based on one productive year only, corresponding to the time the study was carried out or the year for which data was available (Bessou et al., 2013; Cerutti et al., 2013). However, models of annual cropping systems do not account for differences in N cycling that occur during the growth cycle of perennial crops such as oil palm. Some key parameters in these dynamics, such as the length of the crop cycle, the immature and mature stages, inter-annual

yield variations are thus not accounted for. This also applies to other long-term eco-physiological processes, such as the delay between  inflorescence meristem initiation and fruit bunch harvest. To improve the reliability and representativeness of the environmental impacts of oil palm,  we thus 
[revised manuscript text omitted]
 literature references. For emission factors and other parameters,  some ranges were directly provided by some sub-models (e.g.., -IPCC 2006. Other parameters  were varied within a -90% to +90% range ofrelative to their nominal value. The ranges and references are listed in Table SM1 in the Supplementary material. For the analysis, each range was normalised between 0 and 1 and then split into 5 levels by the "morris" function.

The Morris sensitivity analysis technique belongs to the class of "One-at-a-time" sampling designs. For each model, we carried out $400*(n + 1)$ runs, with each set of $n + 1$ runs called a "trajectory". For each trajectory, an initial model run was carried out in which each input variable was randomly set to one of the 5 possible levels. For the second run, one variable $X_1$ was changed to another random level differing from the initial one, and the difference in output between the first and second runs was recorded. That difference, divided by the normalised change in input level, is called an "elementary effect" of variable $X_1$. For the third run, another variable $X_2$ was changed, keeping all other input variable values the same as in the second run. The elementary effect of $X_2$ was recorded, and so on, until the $(n + 1)th$ run. Each trajectory was initiated using a new random set of input variable values, and each trajectory generated one elementary effect value for each $X_1$.

Then, following Morris's method, we calculated two sensitivity indices for each variable $X_i$: the mean of absolute values of the 400 elementary effects ($\mu_i^*$), being the mean influence on the output when the input varies in its min/max range, and their standard deviation ($\sigma_i$). The higher $\mu_i^*$ was, the more influential was the variable $X_i$. The higher $\sigma_i$ was, the more important was the interaction between the variable $X_i$ and the other input variables in the model, or the influence of $X_i$ was nonlinear. The mean of their absolute values of the elementary effect ($\mu_i^*$) was used rather than the mean of the actual values ($\mu_i$), because the effects could be positive or negative.

**3. Results**

**3.1. Comparison of the 11 comprehensive models**

Estimations of total losses of N were very variable, ranging from 21 to 139 kg N.ha$^{-1}$.yr$^{-1}$ around an average of 77 kg N.ha$^{-1}$.yr$^{-1}$ (Figure 1.a). Annual estimates were 20-25 t of fresh fruit bunches.ha$^{-1}$.yr$^{-1}$ for yield, 132-147 kg N.ha$^{-1}$.yr$^{-1}$ of N inputs (mineral fertiliser, atmospheric deposition, biological N fixation, empty fruit bunches and previous felled palms), with 2407 mm.yr$^{-1}$ of rainfall and 932-1545 mm.yr$^{-1}$ of evapotranspiration. Two main factors  contributed to theis variability of N losses: 
[revised manuscript text omitted]
 was apparent a priori in the structures of the models, which were process-based or regression-based, had a yearly or daily time-step, and were more or less comprehensive in terms of processes accounted for. included process-based or regression-based, yearly or daily time-step, and more or less comprehensive models. We may assume that other models exist, which we could not access or calibrate, but those tested very likely provide a representative sample of modelling possibilities for  simulating the N budget in oil palm plantations. Some modelwere clearly operated beyond their validity domains, especially regression-based models for leaching. As this study did not aim to validate the robustness of the models, we did not filter out any of them and the overall set of model outputs helped highlight key fluxes and uncertainties. Further modelling work across contrasting plantation situations might be worthwhile to further test the validity of the models. In particular, nutrient, water or disease stresses or the impact of the previous land use may critically influence the overall crop development and associated N budget.

The variability in model type or structure resulted in large range of model outputs for the oil palm case simulated. There was an approximate 7-fold difference between the lowest and the highest overall N loss estimates. In order to investigate the plausibility of these estimates, we used a simple budget approach. Assuming that soil N content remained constant over the cycle, N inputs would equal N exported in fresh fruit bunches plus the increase in N stock in palms plus N lost. The assumption of constant soil N appears reasonable because soil N dynamics are closely related to soil C dynamics, and soil C stocks in plantations on mineral soil have been shown to be fairly constant over the cycle, especially when oil palm does not replace forest (Smith et al., 2012; Frazão et al., 2013; Khasanah et al., 2015). In our scenario based on measured values (Pardon et al., 2016), average N inputs, N exported and N stored in palms were 156, 60 and 22 kg N.ha$^{-1}$.yr$^{-1}$, respectively. Assuming a constant N stock over the cycle, these values imply N losses of 74 kg N.ha$^{-1}$.yr$^{-1}$.

Based on this plausible estimate of 74 kg N.ha$^{-1}$.yr$^{-1}$, it was possible to identify three groups among comprehensive models: models which likely underestimated losses (IPCC, Mosier, Ecoinvent V3, NUTMON), models which likely overestimated losses (SNOOP, WANULCAS), and models simulating a plausible amount of losses (Banabas, Meier 2014, Brockmann, APSIM, Schmidt).

Underestimates may be due to simulated leaching losses being too low. This was particularly clear for SQCB-NO3 and NUTMON, which used regressions not adapted to the high N uptake rates of oil palm, resulting in . negative leaching losses in some instances. However, IPCC, Mosier and SQCB-NO3, estimated leaching losses within the of 3.5 to 55.8 kg N.ha$^{-1}$.yr$^{-1}$ range of measured losses when considering leaching, runoff and erosion combined  (Figure 4). All models seemed to underestimate NH$_3$ volatilisation compared with measured values (Figure 4). However, this was due to the fact that the higher measured value of 42 kg N.ha$^{-1}$.yr$^{-1}$ was for mineral fertiliser applications of solely urea, whereas the rate of urea in our scenario was 25% of mineral fertiliser. For the IPCC, Mosier and SQCB-NO3 models, the underestimation may also be explained by the fact that none of them were

complete in terms of N budgets. They accounted neither for all gaseous emissions, such as emissions of $N_2$, nor for all inputs, such as atmospheric deposition.

Overestimates of losses were primarily related to leaching losses, which were very high for both WANULCAS and SNOOP. This  could result from  interactions developing between modules in process-based models For instance, the zoning of the palm plantation might have interacted with N inputs in WANULCAS, as the mineral N input from fertiliser was applied close to the palm trunks where water infiltration is likely to be higher due to stemflow. Another potentially important interaction involves N immobilisation and mineralisation in soil. Indeed, in WANULCAS, the mineralisation of residues and empty fruit bunches caused high losses through leaching in the first years of the crop cycle, while in APSIM, the immobilisation of N dominated the dynamics over several years and leaching losses were delayed and reduced to a large extent. However, more work is necessary to better understand how the structure of the model an lead to overestimate leaching.

The models that came up with a plausible estimate of overall N losses, i.e. close to 74 kg N.ha$^{-1}$.yr$^{-1}$, showed large differences in single N flux sizes. APSIM estimated a plausible overall loss of 84 kg N.ha$^{-1}$.yr$^{-1}$, but its prediction of leaching seemed too large compared to measured values. This was very probably because some other fluxes were not taken into account, such as $NH_3$ volatilisation and N input through empty fruit bunches. Similarly, Meier 2014 and Brockmann output plausible overall loss estimates, but large leaching losses  while neither $N_2$ emissions nor N input through biological N fixation were taken into account. Schmidt and Banabas estimates seemed plausible and they accounted for most of the fluxes. Modelled $N_2O$ emissions were similar to field measurements, although the minimum modelled emissions were still higher than the minimum losses measured in the field. Therefore, our results call for caution with regard to the choice of a single model to simulate N losses in oil palm. In absence of further empirical studies available to test these models, we would recommend ing several models to predict N losses.

Some notable patterns differentiated process-based versus regression-based models, and more comprehensive versus less comprehensive models. The process-based models tended to predict higher overall losses and appeared to overestimate leaching losses. The less comprehensive models either seemed to underestimate overall losses, or tended to overestimate leaching losses, which counterbalanced missing fluxes in the N budget. Regarding leaching losses, the process-based models produced similar estimates those that deduced these losses from the total balance.

Process-based models have the advantage of being able to simulate the impact of management practices, such as the timing, splitting and placement of fertilisers. They also take into account other  processes  related to the N cycle, such as carbon cycling, plant growth and water cycling. However such models need more data, e.g. related to soil characteristics Furthermore, the interactions between modules may generate unexpected behaviors, e.g. for simulating leaching, and they are generally not easily handled by non-experts. On the other hand, simple models, such as IPCC and Mosier, have the potential to provide plausible results if some N fluxes were supplemented, without requiring a lot of

data. However they cannot take into account peculiarities of oil palm or the effects of management practices. One way forward is the development of simple models, such as agro-ecological indicators based on the Indigo© concept (Girardin et al., 1999). These indicators are designed to be easy to use while incorporating some specificities of crop system such as management practices.

**4.2. Challenges for modelling the N budget in oil palm plantations**

We identified two important challenges for better modelling the N cycle in oil palm plantations: 1) to model most of the N inputs and losses while accounting for the whole cycle, and 2) to model particular processes more accurately by accounting for the peculiarities of the oil palm system (Table 2).

Given the changes in N dynamics, management practices, and N losses through the growth cycle of oil palm, it is important for models to be built in a way that accounts for this whole cycle. In particular, the immature phase is an important period to consider, as about a third of the N losses occurred during this phase according to the models. Measurements in the field have also shown losses to peak during this phase (Pardon et al., 2016), which involves large inputs of N from the felled palms, the spreading of empty fruit bunchand biological N fixation. This results in complex N dynamics in the understory crop, litter and soil components of the ecosystem. Regarding N input, it seems important to also account for biological N fixation and atmospheric deposition since their contributions to the N budget were not negligible, besides fertiliser applications. Internal fluxes, such as the decomposition of felled palms and residues of oil palm and groundcover, are among the largest fluxes in the oil palm system and their influence on N dynamics is substantial (Pardon et al., 2016). In the case of a new planting, the impacts of land use change and land clearing might also need to be further investigated to better quantify the input fluxes due to decomposition as well as the influence of transitional imbalance state of the agroecosystem on N loss pathways.

For N losses, further model development is also needed to complete the N budget. First, it would be worthwhile to model erosion without requiring detailed input data, while accounting for changes in erosion risk through the crop cycle and the effects of erosion control practices on N dynamics. Erosion was not modelled independently of other losses in most of the reviewed models. Further, $NH_3$ emissions from leaves could easily be included. Finally, despite the difficulties of understanding and simulating the complexity of processes driving $N_2O$ emissions (Butterbach-Bahl et al., 2013), $N_2O$, $NO_x$ and $N_2$ should be modelled in a more comprehensive and systematic way. In particular, $N_2O$ emissions, and thus presumably $NO_x$ and $N_2$ emissions, have high spatial and temporal variability (Ishizuka et al., 2005) Parameters related to fertiliser application are therefore not the only drivers of these emissions, as surmised in the simple models . Since the time resolution of $N_2O$ measurements in the field influence significantly the cumulative emissions recorded for this gas (Bouwman et al. 2002b) it is paramount to model those N losses accounting for the changes in driving parameters over the whole crop cycle.

Finally, losses should not be calculated jointly if the objective is to assess the environmental impacts of the plantation and to identify those practices most likely to reduce N losses and impacts.

Indeed, different N fluxes may lead to different N pollution risks. N losses through erosion, runoff or leaching do not end up in the same environmental compartments, e.g. surface water versus ground water. They hence do not contribute in the same way to potential environmental impacts such as eutrophication. For the purpose of environmental assessment, models should hence be as comprehensive and detailed as possible. Regarding these criteria, the Schmidt model appeared the most comprehensive and detailed one, as it distinguishes between six N fluxes. However, this model could be improved by modelling separately losses through erosion, runoff and leaching, i.e. calculating a total of eight N fluxes.

The second challenge is to improve the model some of the key N cycling processes while accounting for the peculiarities of the oil palm system. Regarding internal fluxes, a better representation of the interaction between legumes and soil N dynamics is an important challenge, as the actual role of legumes during the immature period is complex and not fully understood yet. Indeed, legumes have the capacity to regulate their N provision, by fostering N fixation or N uptake, depending on soil nitrate content (Pipai, 2014; Giller and Fairhurst, 2003). They may contribute to the reduction of N losses through immobilisation or to their increase through N fixation and release.

Reducing the uncertainty in the modelling of leaching  is an important challenge, as about 80% of the total losses came from leaching, according to the models, and results were very variable across models. Models should be better adapted to the oil palm system, as some regression models clearly appeared out of their validity domain. Further research on leaching prediction should focus on the effects of soil clay content, oil palm rooting depth and oil palm N uptake, since they emerged as he most influential variables according to the sensitivity analysis. The  -90% to +90% relative variation range used in the latter for the parameters which were not given a specific range may appear as a rather extreme set of values, but they made it possible to encompass a wide range of conditions. The sub-models included in the sensitivity analysis were regression models which did not explicitly simulate N cycling processes, resulting in a lack of influence of some parameters which may affect leaching in practice and in process-based models.  Therefore, it could be interesting to perform complementary sensitivity analyses focused on process-based models, such as APSIM.

In order to take into account the influence of management practices on internal fluxes and losses, it would be necessary to use a daily step approach, to account for the timing or splitting of N fertiliser applications. Modelling approaches that incorporate spatial heterogeneity, as in WANULCAS, should be favoured. to assess the effect of fertiliser or empty fruit bunch placements. For gaseous losses, emission factors could be adapted to the oil palm system, as all of them, i.e. for $NH_3$ or $N_2O/NO_x$ fluxes, were based on data from temperate areas on mineral soils, including mostly animal manure as reference for organic fertilisers. On a general note, more

field measurements and model development are needed to account for the peculiarities of palm plantation management on peat soil. They involve  substantial and potentially widespread, notably in Indonesia (Austin et al., 2015). Those plantations require specific management, including complex drainage systems, and may entail severe pollution risks, notably leaching, that are not yet properly accounted for in current models, e.g. IPCC 2006.

**TABLE 2. about here**

**4.3. Implications for management**

The main levers managers can use to reduce N losses involve the level of  input, including fertiliser management, but also the handling of the immature phase. To manage fertiliser inputs, managers need to know economic response, which is the main driver of practices, and environmental response, to type, rate, timing and placement. They may decide on the optimum fertiliser management practices based on these two dimensions. Models that include both N losses and fresh fruit bunch production in relation to  management scenarios can provide the information needed to evaluate both responses.

The model comparison showed the importance of the immature phase with respect to N losses, and suggested field research lines and modelling approaches to improve our understanding of loss processes and their estimat. There are also direct implications of our results for crop management during this phase. Light, water and N are not  fully used by the young palms, as their canopies and root systems do not completely exploit the field. Thus, in the current systems, the combination of high input rates with sub-optimal  resource capture capacity of  the growing oil palms in the immature period results in high losses and negative environmental impacts. There are two possible approaches for reducing theose. One is to reduce the inputs: for instance, it might be better to plant a non-legume cover crop and to manage N supply to the palms only with fertiliser. An alternative approach would be to grow another crop during this phase, which would use the surplus N and either export it in product or take it up in biomass that would decompose later. For instance, fast growing trees like balsa; trunks could be harvested after 5 years and exported, whilst leaving some branches, leaves and roots to decompose on the soil.

There also exist re-planting systems  that make it possible to combine old and young palm trees in the same plantation block. The advantage can be both economic and agro-ecological as the immature phase actually becomes productive thanks to the remaining old palm trees and the nutrient cycling potentially more competitive. However, there is still limited data available to quantify and model the potential competition and adapt fertiliser management. Moreover, potential reduction in N losses should not come at the cost of increased use  of herbicides, which may be used to kill the old palm trees without damaging the newly planted ones.

From the environmental point of view, it is also important to consider fertiliser management and N losses within a wider system and value-chain. First, fertilisers encompass residues from the mill, whose environmental cost and /benefit to the plantation should be  considered from a whole life cycle perspective. This would include the production of waste, transport, or avoided

impact through the substitution of synthetic fertilisers, etc. This can be done using life cycle assessments. Second, the carbon balance, i.e. the balance of carbon sequestration and release, is closely coupled to the N balance. Thus models that include both cycles are warranted to fully evaluate the environmental impacts of oil palm production.

**5. Conclusion**

N losses are a major concern for when assessing the environmental impacts of oil palm cultivation, and management practice targeted at reduceing N losses and costs is critical to this industry. Modelling N losses is crucial because it is the only feasible way to predict the type and magnitude of losses, and thus to assess how improved management practices might reduce losses. Our study showed that there were considerable differences between existing models, in terms of model structure, comprehensiveness and outputs. The models that generate N loss estimates closest to reality were the most comprehensive ones, and also took into account the main oil palm peculiarities, irrespective of their calculation time step. However, in order to be useful for managers, a precise modelling of the impact of management practices on all forms of N losses seems to require the use of a daily time step or the modelling of spatial heterogeneity within palm plantations. The main challenges are to better understand and model losses through leaching, and to account for most of the N inputs and outputs. Leaching is the main loss pathway and is likely to be  high during the  young phase when inputs are high due to decomposition of felled palms and N fixation by legumes. Field Ddata are still needed to better understand temporal and spatial variability of other losses as well, such as $N_2$, $N_2O$ and $NO_x$ emissions, in the context of oil palm. These improvements could allow managers to evaluate the economic and environmental impacts of changes in management, such as for instance, modifying fertiliser inputs or the plant cover type during the immature phase.

**Acknowledgements**

[revised manuscript text omitted]
